# Developmental kinetics and transcriptome dynamics of stem cell specification in the spermatogenic lineage

Nathan C. Law[1], Melissa J. Oatley[1] & Jon M. Oatley [1]

Continuity, robustness, and regeneration of cell lineages relies on stem cell pools that are established during development. For the mammalian spermatogenic lineage, a foundational spermatogonial stem cell (SSC) pool arises from prospermatogonial precursors during neonatal life via mechanisms that remain undefined. Here, we mapped the kinetics of this process in vivo using a multi-transgenic reporter mouse model, in silico with single-cell RNA sequencing, and functionally with transplantation analyses to define the SSC trajectory from prospermatogonia. Outcomes revealed that a heterogeneous prospermatogonial population undergoes dynamic changes during late fetal and neonatal development. Differential transcriptome profiles predicted divergent developmental trajectories from fetal prospermatogonia to descendant postnatal spermatogonia. Furthermore, transplantation analyses demonstrated that a defined subset of fetal prospermatogonia is fated to function as SSCs. Collectively, these findings suggest that SSC fate is preprogrammed within a subset of fetal prospermatogonia prior to building of the foundational pool during early neonatal development.

---

[1] Center for Reproductive Biology, School of Molecular Biosciences, College of Veterinary Medicine, Washington State University, Pullman, WA 99164, USA. Correspondence and requests for materials should be addressed to J.M.O. (email: joatley@wsu.edu)

R are populations of stem cells maintain cell lineages through a delicate balance between production of transit amplifying progenitors and maintenance of the stem cell reservoir via self-renewal. In general, the mechanisms that drive specification of stem cells among the remaining lineage during development remain largely unknown. For the mammalian spermatogenic lineage, spermatogonial stem cells (SSCs) are the foundation from which males continually produce millions of genetically unique gametes daily from puberty until old age[1]. Impaired establishment or maintenance of the SSC pool can manifest as either germline ablation or development of testicular seminomas[2,3].

In mice, the most well-studied mammalian model of germline development, the origins of the spermatogenic lineage can be traced to primordial germ cells (PGCs) that emerge from the proximal epiblast around embryonic (E) 6.25[4]. PGCs migrate to the developing gonad, undergo sex determination, and if XY, transition to form prospermatogonia[5–7], the immediate precursors to the entire male spermatogenic lineage including SSCs (Fig. 1a). Following sex determination, prospermatogonia proliferate before entering a phase of quiescence around E16.5, at which point DNA methylation patterns are reestablished. After birth, germ cells progressively migrate from the center of

**Fig. 1** Germ cell subsets emerge from a pool of quiescent prospermatogonia. **a** Schematic of germline development in males. **b** and **c** Transgenic mouse model (**b**) and representative images (**c**; scale bar, 200 μM) for tdTomato fluorescent labeling of the male mouse germline via *Blimp1-Cre* and in combination with *Id4-eGfp* transgene expression. **d** Representative images from whole-mount seminiferous cords of ID4-eGFP expression starting in E16.5 prospermatogonia (scale bar, 40 μM). White dotted lines approximate seminiferous tubule borders and numbers identify clustering of eGFP+ cells. **e** Flow cytometric analysis (FCA) of eGFP+ and total germ cells. **f–g** FCA cell cycle analysis (**f**) and distribution (**g**) of ID4-eGFP− and ID4-eGFP+ populations. **h** Quantification of ID4-eGFP distribution divided into Bright, Mid, and Dim subsets. **i–l** t-Distributed stochastic neighbor embedding (tSNE) representation of gene expression for select pluripotency markers, *Dppa3* (**i**), *Sox2* (**j**), *Nanog* (**k**), and *Pou5f1/Oct4* (**l**), from scRNA-seq analysis of E16.5 germ cells. **m** and **n** Graph-based clustering results (**m**) and associated heatmap of select differentially expressed genes (**n**) from E16.5 prospermatogonia. FCA data in **e–h** are gated from all tdTomato+ cells from isolated gonads (sample gating procedure located in Supplementary Fig. 2d). Quantifications in **e–h** are presented as means with error bars representing SEM for n = 3–5 biologically independent animals per age point (n = 3 for E16.5 and E18.5; n = 4 for P1 and P2; and n = 5 for P0). Source data are provided as a Source Data file. Transcriptome scRNA-seq analysis in **i–n** are representative of 3845 cells from n = 3 biologically independent animals

seminiferous cords to the basal lamina and reenter the cell cycle in the first days of postnatal life[8–10]. The transition from prospermatogonial precursor to postnatal spermatogonia, including SSCs, is thought to occur during a broadly defined timeframe of postnatal days (P) 0–6[11,12].

Several studies have revealed that significant heterogeneity exists in the neonatal prospermatogonia population[11–15]. Histomorphological studies by Kluin and de Rooij[11] first characterized two populations of germ cells in the postnatal testis: one that formed undifferentiated spermatogonia, including presumptive SSCs, and another that transitioned directly to a differentiating state. Genetic studies by Yoshida et al. (2004), revealed that initial differentiating spermatogonia contributing to a first round of spermatogenesis are negative for Neurog3 and subsequent rounds of differentiating spermatogonia are derived from Neurog3 positive progenitors emanating from the SSC pool[12]. Thus, these studies and others suggest that both SSCs and initial differentiating spermatogonia are derived from a seemingly homogenous prospermatogonial population. Contrastingly, recent studies indicated that germ cell heterogeneity is evident in late fetal prospermatogonia[15]. However, little is known regarding the genesis of germ cell heterogeneity in fate specification. Furthermore, the timing, kinetics, and pathways for which the foundational SSC pool is set aside from the remaining germ cell population are undefined. To date, three predominant mechanisms for the specification of SSCs have been proposed[12,16], including (1) stochastic selection from a homogeneous population; (2) preprogramming at an early stage in development; or (3) selective determination based on unknown mechanisms.

A roadblock to defining how and when the postnatal spermatogonial populations, including the foundational SSC pool, arise during development has been the lack of tools to clearly discern SSCs and prospermatogonia fated to become them. Previous studies established that SSCs in mice are marked by expression of the transcriptional repressor inhibitor of DNA binding 4 (ID4)[17–19], and ID4 is functionally important for maintenance of the SSC reservoir[19]. Using an Id4-eGfp transgenic reporter mouse line[18], we determined that the brightest eGFP-expressing spermatogonia (ID4-eGFP^Bright) encompass the SSC-derived regenerative capacity in the germline[20], express hallmark SSC genes[20], and are functionally resistant to retinoic acid (RA)-induced terminal differentiation[21]; this population is denoted as SSC^Ultimate[20–23]. Spermatogonia with lower eGFP expression, classified as ID4-eGFP^Mid and ID4-eGFP^Dim, phenotypically comprise populations transitioning from an SSC to progenitor state and are responsive to signaling by RA[20–23]. How these undifferentiated spermatogonial subsets arise in development has not been explored.

Here, we provide evidence that suggests SSC fate is restricted to a subset of preprogrammed prospermatogonia during fetal development. Core SSC regulators, identified in vivo using transgene expression and in silico using single-cell RNA-sequencing (scRNA-seq), arrange along a continuum and mark subpopulations of fetal and neonatal germ cells. The level of expression for core SSC regulators define populations fated to become SSCs, progenitors, or differentiating germ cells in the postnatal testis. Furthermore, upon mitotic reactivation of the entire germline, SSCs rapidly self-renew before reaching an upper limit, at which point layers of transitioning and differentiating spermatogonia then arise. Moreover, using marker gene expression, we identified SSC-fated subpopulations through development and mapped the transcriptional dynamics underlying the process. Lastly, transplantation analyses with defined subsets of prospermatogonia indicated that SSC fate is functionally preprogrammed in late fetal development.

## Results

**SSC specification during late fetal development.** To track the emergence of SSCs and other germ cell populations in vivo during development, we generated a quadruple-transgenic hybrid reporter mouse model expressing Id4-eGfp and Blimp1-Cre transgenes along with tdTomato^flox_STOP_flox and LacZ in separate Rosa26 alleles (Fig. 1b, c; breeding scheme is described in the section "Methods"). Blimp1-Cre is expressed by PGCs[4]; thus, CRE-mediated activation of tdTomato irreversibly labels descendent germ cells, including prospermatogonia and postnatal spermatogonial subtypes. Assessment of testes by fluorescent microscopy revealed ID4-eGFP expression in a rare subset of prospermatogonia at E16.5 (Fig. 1d). Using flow cytometric analysis (FCA), we quantified the incidence of ID4-eGFP+ germ cells through developmental time. From E16.5 to P2, the total number of ID4-eGFP+ prospermatogonia increased in association with an increase of total prospermatogonia and rise in proliferative index (Fig. 1e, f, and Supplementary Fig. 1a–e). By P1, ID4-eGFP+ prospermatogonia outnumbered their ID4-eGFP− counterparts (Fig. 1g). Furthermore, as increasing numbers of prospermatogonia expressed ID4-eGFP, a gradient emerged in which Bright, Mid, and Dim subsets were discernable beginning at P0 (Fig. 1h and Supplementary Fig. 2a). Taken together, these data indicate that significant heterogeneity exists within the prospermatogonial population.

To explore SSC fate specification further, we performed scRNA-seq analysis of the entire germ cell population isolated from E16.5 testes. After filtering and quality control, a total of 3845 germ cells were isolated from three different mice. Individual transcriptomes were sequenced at an average depth of 130,286 reads per cell (representing 61.2% average sequencing saturation), which captured 20,411 mean genes per library and 28,057 median unique molecular identifier (UMI) counts per cell. Replicate libraries were highly consistent, with modified Pearson's correlations[24] of 0.96–0.99. Preliminary examination of scRNA-seq data revealed considerable heterogeneity amongst fetal prospermatogonia consistent with outcomes of in vivo imaging and FCA. Germline pluripotency markers expressed in founder PGCs and normally downregulated prior to birth, including Dppa3, Nanog, and Sox2[25–28], arranged in a gradient when visualized by t-distributed stochastic neighbor embedding (tSNE) (Fig. 1i–k), indicating that downregulation of the core germ cell pluripotency network occurs asynchronously during development. Interestingly, Pou5f1 (Oct4) expression remained evenly distributed among the population (Fig. 1l), likely reflecting maintenance of Pou5f1 expression into postnatal life.

To determine whether the observed heterogeneity coincides with germ cell fate, we first performed graph-based clustering to subdivide the E16.5 population and generated three unique clusters (Fig. 1m). We then performed differential gene expression (DEG) analysis comparing the cluster with highest Dppa3, Nanog, and Sox2 expression (E16.5 Cluster 1) to all other clusters. A total of 838 significant DEGs (p-value < 0.01) were identified, among which several markers of the postnatal SSC population were upregulated in E16.5 Cluster 1, such as Etv5, Id4, Lhx1, Nanos2, and Ret[20,29–31] (Fig. 1n). Conversely, markers of differentiating progenitors in the postnatal testis, including Dnmt3a, Dnmt3b, Sohlh1, and Sox3[32–34], were significantly downregulated in E16.5 Cluster 1. Thus, expression of core SSC factors is upregulated in a subpopulation of prospermatogonia that retain the germline pluripotency network further along the developmental trajectory.

**SSC pool establishment and postnatal proliferation dynamics.** Previous studies indicate that mouse prospermatogonia reenter

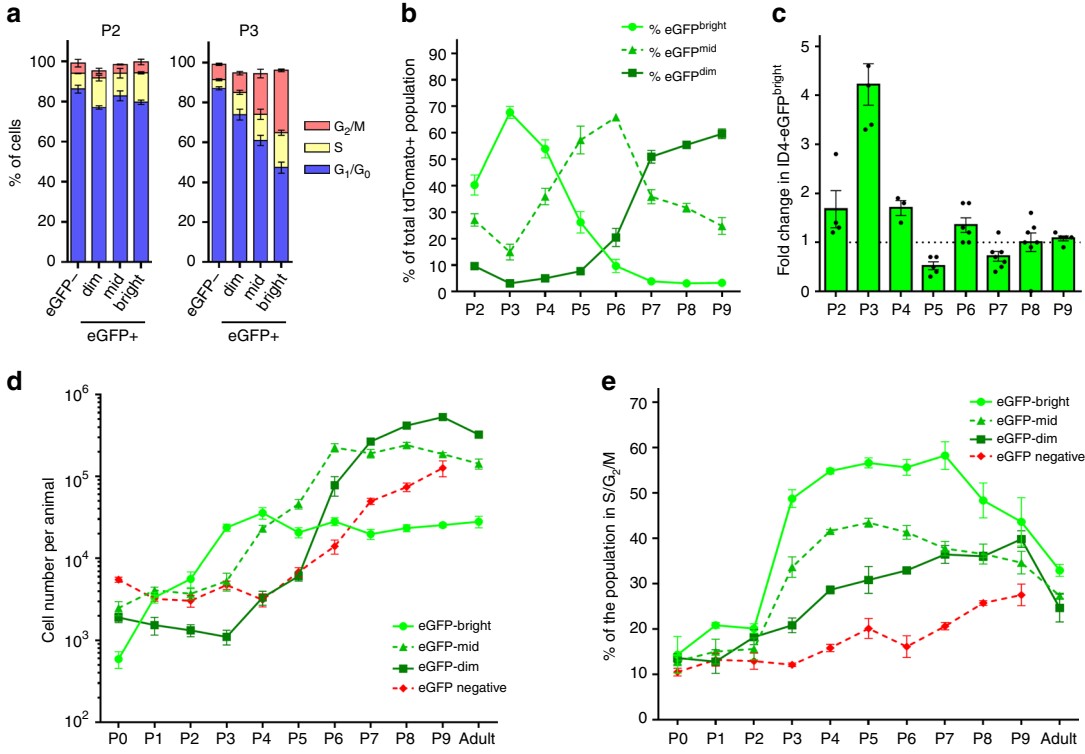

**Fig. 2** Postnatal germ cell mitotic reactivation initiates SSC pool establishment. **a** Cell cycle analysis of germ cell subtypes at P2 and P3. **b** Percent germline distribution of ID4-eGFP+ subsets. **c** Fold-change in number of ID4-eGFP$^{Bright}$ germ cells from the previous developmental day. **d** and **e** Cell number per animal (**d**) and cell cycle status (**e**) during development. Data in **a–e** are presented as means with error bars representing SEM for $n = 3$–7 biologically independent animals at each age point ($n = 3$ for all cell cycle data; otherwise $n = 4$ for P1, P2, P4, and adult; $n = 5$ for P0, P3, P5, and P9; $n = 6$ for P6 and P8; $n = 7$ for P7). Source data are provided as a Source Data file

the cell cycle sometime between P1 and P3[9,10]. We observed an increase in proliferative index among all prospermatogonia beginning at birth (Fig. 1f) and found that germ cell subsets cycled unequally during the first week of postnatal life primarily beginning at P3 (Fig. 2a). The ID4-eGFP$^{Bright}$ subpopulation, which functional transplantation studies have demonstrated represents the SSC pool, cycled fastest leading to a spike in the proportion of the entire germ cell population that was ID4-eGFP$^{Bright}$ at P3 (Fig. 2b). From P2 to P3, the number of ID4-eGFP$^{Bright}$ increased by $4.21 \pm 0.43$ fold ($p < 0.01$ by one-way ANOVA for $n = 5$ biologically independent animals; reported as mean $\pm$ SEM) (Fig. 2c), demonstrating rapid turnover. Interestingly, following a quick increase to ~25,000 cells per animal at P3 (Supplementary Fig. 1f), the number of ID4-eGFP$^{Bright}$ germ cells plateaued and remained constant into adulthood (Fig. 2d). Thus, based on the dynamics of the ID4-eGFP$^{Bright}$ population, establishment of the foundational SSC population is triggered at P0 and completes by P3.

Importantly, while cell number plateaued by P3, ID4-eGFP$^{Bright}$ cells remained actively cycling for several days as evidenced by the high percentage of cells in S/G$_2$/M (Fig. 2e), suggesting a switch from self-renewal to the production of ID4-eGFP$^{Mid}$ cells that are in transition to a progenitor state. Consistent with this shift, the proportion of ID4-eGFP$^{Mid}$ germ cells steadily increased from P3 to P6 (Fig. 2b), followed by another plateau in cell number that persisted into adulthood (Fig. 2d). Finally, shortly before the ID4-eGFP$^{Mid}$ peak, the proportion of ID4-eGFP$^{Dim}$ cells grew until reaching an upper limit between P8 and P9 that also remained into adulthood (Fig. 2b, d). Expansion of the ID4-eGFP− population trailed all ID4-eGFP+ subtypes, but steadily propagated (Fig. 2d) despite remaining a minor subset of the entire germ cell population

between P3 and P9 (Supplementary Fig. 2c). Together, these data illustrate that following establishment of the SSC population, assembly of the spermatogenic lineage occurs in a top-down fashion with the production of sequential layers of germ cells transitioning to a differentiating state.

Further evaluation of the proliferation kinetics data overall revealed that cell cycle dynamics of the germ cell populations during neonatal development followed divergent paths (Fig. 2e). The ID4-eGFP$^{Bright}$ and ID4-eGFP$^{Mid}$ populations entered a phase of early rapid cell cycle progression starting at P3 that eventually trailed off through P9. Contrastingly, ID4-eGFP$^{Dim}$ and ID4-eGFP− populations slowly increased in proliferative index into P9. Finally, all adult ID4-eGFP+ populations cycled at relatively equal rates significantly greater than those of the prospermatogonial precursors. While the presence of slow-cycling SSCs in the adult testis remains contentious[1], these data suggest that adult SSC$^{Ultimate}$ and transitory spermatogonial populations are equally proliferative but not necessarily quiescent.

**Male germ cells arrange in nests during development.** During initial imaging of testes from quadruple-transgenic mice (Fig. 1d), we observed that ID4-eGFP+ germ cells localized to distinct regions along seminiferous cords/tubules. Within these regions, ID4-eGFP+ germ cells were located either adjacent to one another (Fig. 1d, cells 1 and 2), being possibly connected via an intercellular bridge, or were separated by short distances and lacked apparent intercellular connection (Fig. 1d, cells 2 and 3). This localized clustering of germ cells resembled nests similar to what has been observed in the fetal ovary and testis around E13.5[35–37]. To explore this further, we utilized 3D confocal

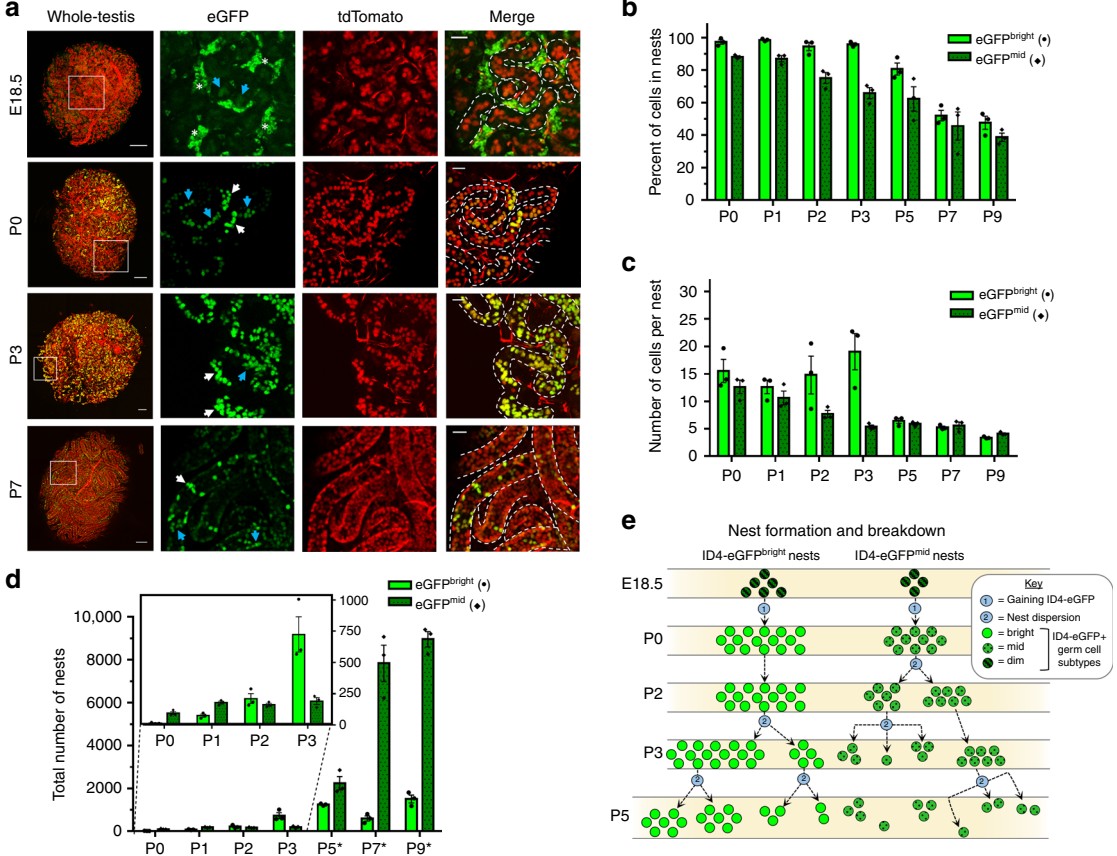

**Fig. 3** ID4-eGFP germ cells form nests during fetal and neonatal development. **a** Whole testes were imaged by 3D confocal microscopy. Select confocal slices from complete testis Z-stacks (Whole-Testis panels, white square) illustrate examples of ID4-eGFP[Bright] (eGFP panels, white arrows) and ID4-eGFP[Mid] (eGFP panels, blue arrows) nests among all tdTomato+ germ cells. Seminiferous tubule borders are outlined (white dotted lines) and embryonic interstitial Leydig cell autofluorescence is indicated (E18.5, eGFP, white asterisks); eGFP, tdTomato, and Merged panels share equal scales (scale bar, 50 μM) different from Whole-Testis panels (scale bar, 200 μM). **b–d** Percent of cells in nests (**b**), average nest size (**c**), and total number of nests (**d**) were quantified from 3D confocal images for ID4-eGFP[Bright] and ID4-eGFP[Mid] populations. Total number of nests for P5, P7, and P9 were estimated as outlined in the section "Methods" (**d**, asterisks). Data are presented as means with error bars representing SEM for $n = 3$ biologically independent animals at each age point. Source data are provided as a Source Data file. **e** Schematic portraying eGFP+ germ cell nest dynamics. Numbering key indicates different actions based on 3D confocal analyses

microscopy of whole mouse testes (Fig. 3a) and quantified the size, frequency, and incidence of germ cell nests during development. We defined a nest as a grouping of three or more germ cells separated by no more than 10 μm from each other; for reference, the average measured germ cell diameter was ~20 μm. Only cell clusters containing ≥3 germ cells were considered nests based on the rationale that two cells could be in close proximity by chance or following cell division but prior to migration away from each other.

Outcomes of 3D imaging revealed that during late fetal development, nests with dim ID4-eGFP intensity were evident (Fig. 3a, E18.5, blue arrows). However, Leydig cell autofluorescence obstructed nest quantification of fetal testes (Fig. 3a, E18.5, white asterisks). Quantification of P0–P3 testes revealed that nearly all ID4-eGFP[Bright] and most (>75%) ID4-eGFP[Mid] germ cells were present in nests (Fig. 3b). From P0 to P2, the size of ID4-eGFP[Bright] nests remained relatively constant (Fig. 3c), but the total number of nests steadily increased (Fig. 3d, inset), suggesting that germ cells gained ID4 expression in localized regions along seminiferous tubules (depicted in Fig. 3e, identified with 1). By P2, ~210 ID4-eGFP[Bright] nests were present throughout the entire testis.

Interestingly, germ cell nests were not sustained throughout development; nests dispersed asynchronously depending on the ID4-eGFP subset. The average nest size of ID4-eGFP[Bright] cells did not significantly increase during the proliferative response at P3, but the total number of nests expanded, indicating the start of nest breakdown for ID4-eGFP[Bright] germ cells (Fig. 3e, identified with 2). The percentage of nested ID4-eGFP[Bright] germ cells steadily declined and the number of cells per nest dropped in subsequent proliferative days (P5–P9), illustrating continued nest breakdown of ID4-eGFP[Bright] germ cells. In contrast, dispersion of ID4-eGFP[Mid] nests commenced around P1–P2, at which point the percentage nested and average nest size began to decline. While ID4-eGFP[Mid] nest size remained relatively constant from P5 to P9, the total number of nests increased consistent with population expansion. Taken together, germ cell nest dynamics suggests that spatiotemporal clustering plays a key role in fate determination of the different spermatogonial subtypes during development.

**Transcriptome profiles identify preprogramming of SSC fate.** To further define the developmental trajectory of SSCs and their prospermatogonial precursors, we performed additional scRNA-seq analysis on germ cells isolated from P0, P3, and P6 testes. When merged with the E16.5 prospermatogonial transcriptome dataset, a total of 10,140 germ cells were isolated from eight

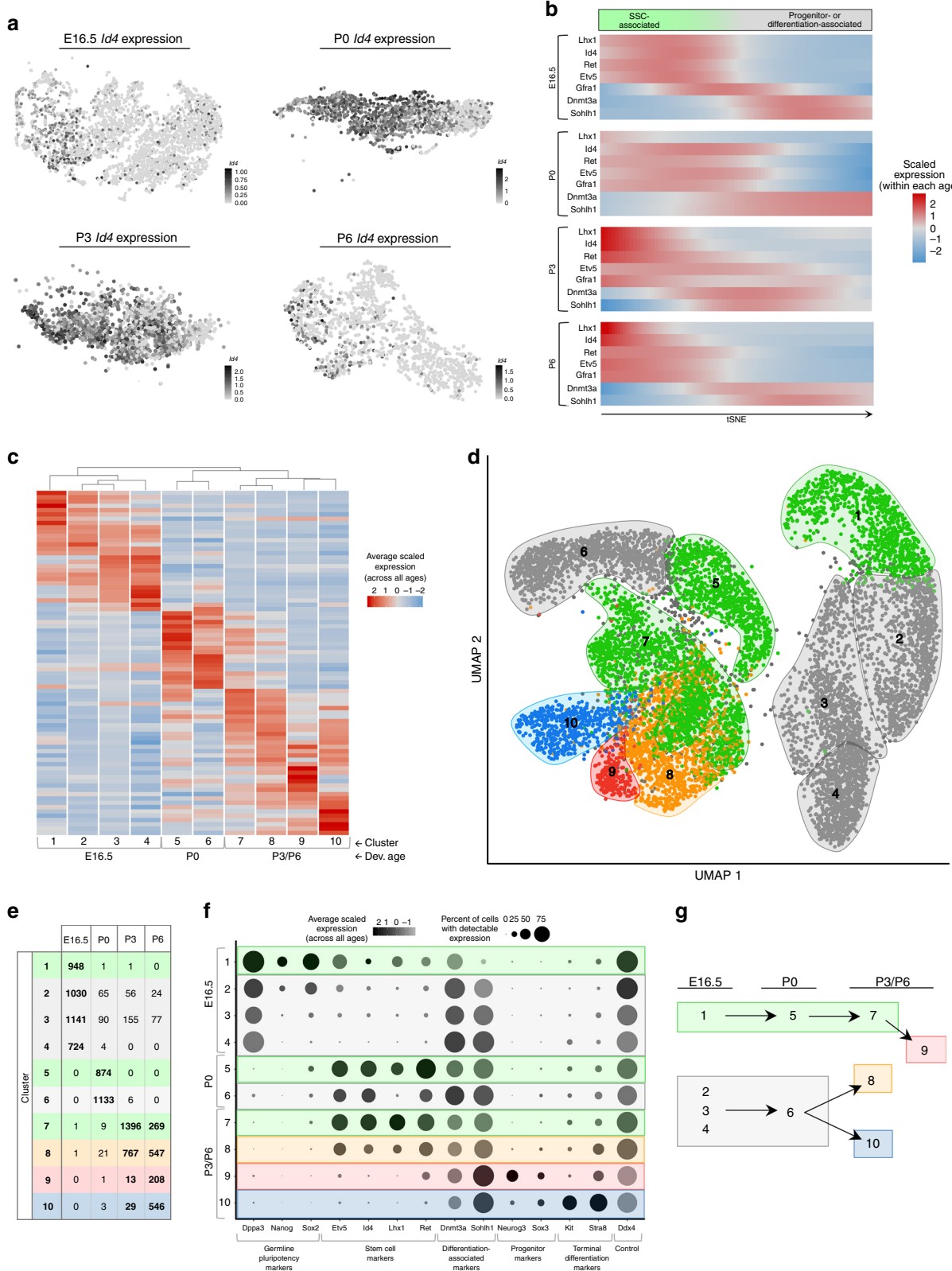

different mice across the four developmental ages. Individual transcriptomes were sequenced at an average depth of 127,675 reads per cell representing moderate sequencing saturation (62.6% average sequencing saturation). Together, the pooled libraries detected 20,325 mean genes per library and 20,546 median UMI counts per cell. Modified Pearson's correlations of

0.96–0.99 among replicates within each developmental age indicated highly consistent datasets.

Initial gene expression analysis of each developmental age by tSNE distribution revealed heterogeneous *Id4* expression consistent with in vivo transgene expression (Fig. 4a). Therefore, based on the presence of population heterogeneity through

**Fig. 4** Mapping the SSC trajectory through development with graph-based clustering. Single-cell RNA-sequencing was performed on germ cells isolated from P0, P3, and P6 gonads and merged with E16.5 transcriptome data presented above. **a** tSNE representation of *Id4* expression heterogeneity present at each developmental age, consistent with transgene expression analysis above. **b** Cells from each developmental age were ordered in an unbiased manner via one-dimensional tSNE or pseudotime. Subsequent heatmap representation illustrates that expression of SSC-associated genes, such as *Lhx1, Id4, Ret, Etv5,* and *Gfra1* arranged opposite differentiation-associated factors, such as *Dnmt3a* and *Sohlh1* along each continuum. **c** Heatmap of the top 10 differentially expressed genes between 10 graph-based clusters generated in the aggregate dataset. Genes are listed in Supplementary Data 1. **d** Uniform manifold approximation and projection (UMAP) distribution of the 10 graph-based clusters. **e** Table of age distribution for each cluster. Bold numbering indicates age most represented within each cluster. **f** Dotplot representation of average scaled expression (across all ages, color gradient) and the percentage of cells within each cluster with detectable expression (dot radius) for select marker genes that identify each population. **g** Trajectory estimates for each graph-based cluster based on marker gene expression and cell number distribution. Single-cell RNA-seq data from **a–f** are representative of a total of 10,140 cells from $n = 8$ biologically independent animals

development, we first performed trajectory inference algorithms such as Monocle, Wishbone, destiny, URD, etc. to estimate the presence of distinguishing lineages. Unfortunately, these models were either unable to resolve unique trajectories (Supplementary Fig. 3a, b) or incorrectly ordered developmental ages along a trajectory due to complexities within the dataset. Further evaluation of each developmental age revealed two patterns of heterogeneity: (1) a gene expression continuum within each developmental age and (2) population differences through developmental time. Interestingly, with the cells of each developmental age ordered along a continuum based on transcriptomic signature via one-dimensional tSNE or pseudotime (Fig. 4b), we found that SSC-associated genes, such as *Lhx1, Id4, Ret, Etv5,* and *Gfra1*[19,29,30,38] generally organized along the continuum opposite differentiation-associated genes, such as *Dnmt3a* and *Sohlh1*[32,34]. Due to this complexity, trajectory inference analysis commonly placed SSC-associated or progenitor/differentiation-associated regions together in a head-to-tail fashion due to transcriptional similarities among the associated regions at each developmental age (depicted in Supplementary Fig. 3c). As a result of these limitations, we employed a graph-based clustering approach to identify SSCs and their prospermatogonial precursors. We hypothesized that in a preprogramming model of SSC fate specification, a subset of germ cells with a distinguishing transcriptomic signature could be identified across multiple ages. Furthermore, as a subset of germ cells upregulated key SSC factors in each developmental age, the trajectory of this population could be tracked through developmental time.

We utilized a popular clustering method which involves generating a *K*-nearest neighbor graph based on Euclidean distance in principal component space followed by the Louvain modularity algorithm to group cells[39,40]. With this approach, parameters were applied to the entire dataset to generate unbiased clusters enriched for germ cell subpopulations based on transcriptomic signatures and functional evidence supporting cell identity. Our previous studies demonstrated that the ID4-eGFP^Bright population at P6 represents a quantifiably pure SSC population[20] and findings presented in Supplementary Fig. 1i indicate that the ID4-eGFP^Bright population constitutes ~10% of the entire germ cell population at P6. Therefore, we selected a global clustering resolution such that the P6 SSC population was enriched within a single cluster. From this global clustering analysis, 10 unique clusters were generated (Fig. 4c, d, and Supplementary Data 1), including cluster 7 which was enriched for P6 SSCs based on marker gene expression (Supplementary Fig. 4b) and relative distribution within P6 (Supplementary Fig. 4c).

Based on analyses above and previous studies[20,41,42], the relative expression level for select marker genes along a continuum defines SSCs and more differentiated progenitors. Currently, the SSC population cannot be isolated in purity by a single marker. Thus, we used the age distribution of cells in each cluster (Fig. 4e) and marker gene expression (Fig. 4f), measured

as scaled expression and the percentage of cells with detectable expression for each marker, to assign identities to each cluster relative to each developmental age. In general, marker genes were selected based on scaled expression >0 and transcripts detected in >10% of cells within at least one cluster. SSC-associated and differentiation-associated markers were selected within additional criteria, including functional evidence tying each marker to either SSC function or male germline differentiation and expression across all developmental ages tested (scaled expression >0 and expression detected in >10% of cells within at least one cluster at each developmental age). Cells fated for terminal differentiation, defined by expression of *Kit* and *Stra8*, and progenitor formation from the SSC pool, defined by expression of *Neurog3* and *Sox3*, both occur after P3[14,33,43–47]; thus, expression of these markers is primarily pertinent to P3 and P6 developmental ages. Additionally, germline pluripotency markers, including *Dppa3, Nanog,* and *Sox2*, are downregulated in postnatal development[25–28]; thus, expression of these markers is pertinent to E16.5. Markers that fall within these criteria are presented in Fig. 4f and additional markers that do not are presented in Supplementary Fig. 4d.

First, focusing on E16.5 which was represented primarily by clusters 1–4, cluster 1 possessed the highest average expression levels of SSC markers *Etv5, Id4, Lhx1,* and *Ret*[19,29,30], and the lowest levels of the differentiation-associated markers *Dnmt3a* and *Sohlh1*[32,34]. Furthermore, cluster 1 overlapped by 82.9% with E16.5 Cluster 1 in Fig. 1 and possessed the highest levels of germline pluripotency markers[28] (*Dppa3, Nanog,* and *Sox2*). Second, P0 was primarily represented by clusters 5 and 6, and cluster 5 possessed the highest expression of SSC-associated markers and lower expression of differentiation-associated markers. Interestingly, while SSC-associated and differentiation-associated markers still arranged in a relative continuum at P0, *Lhx1* represented the most restrictive marker. Third, P3 and P6 datasets shared significant overlap among the remaining clusters, likely reflecting similar population-based transcriptomic signatures established after SSC pool formation at P3; thus, the two ages were analyzed together. Among clusters represented at P3 and P6, cluster 7 possessed the highest average expression of SSC-associated markers and distributed among the entire germ cell populations at both P3 and P6 comparable to the ID4-eGFP^Bright population measured by FCA (Fig. 2). Taken together, we concluded that clusters 1, 5, and 7 represent SSCs at P3–6 and prospermatogonia programmed to become them at E16.5 and P0 (Fig. 4g).

Beyond defining the SSC trajectory, we were also able to infer the formation of other germ cell subpopulations using marker genes and cell distribution among the developmental ages. Based on upregulation of *Kit* and *Stra8*, a population comprising the first round of differentiating spermatogonia in the postnatal testis was seemingly identified (cluster 10). To confirm this, we evaluated the KIT expression phenotype by germ cell subsets at P3. Outcomes of FCA revealed that ~12% of the total germ cell population is KIT+ at this developmental age point

(Supplementary Fig. 4e). The makeup of this population was found to be primarily (~93%) ID4-eGFP<sup>Dim</sup> and ID4-eGFP− germ cell subsets; whereas, minor portions could be classified as ID4-eGFP<sup>Mid</sup> (~6%) or ID4-eGFP<sup>Bright</sup> (~1%). These findings aligned with the clustering predictions. Additionally, based on differential expression of *Neurog3* and *Sox3*, an initial round of SSC-derived progenitors was captured by cluster 9. Lastly, cluster 8 encompassed a population of germ cells with lower levels of SSC-associated markers compared to the SSC pool (cluster 7) but lacked markers associated with progenitor (e.g. *Neurog3* and *Sox3*) or differentiating (e.g. *Kit* and *Stra8*) spermatogonial states. We postulated that this cluster represents an initial population of progenitors derived from prospermatogonia that are not destined for the SSC pool but have yet to enter the terminal differentiation pathway.

**Validation of SSC fate association markers by immunostaining.** To assess whether the fate specification predictions generated by scRNA-seq analysis are reflected at the protein level, we chose markers with differential expression (Supplementary Data 1) in clusters 1 (DPPA5A) or 5 (VPS8) that predict subsets of prospermatogonia at E16.5 and P0 that are fated to become SSCs, respectively. In addition, we chose a differentially expressed gene (Supplementary Data 1) in cluster 7 at P6 (HHEX) that predicts an SSC state. Immunofluorescent staining for all three markers revealed differential expression in germ cells that express ID4-eGFP at each developmental age commensurate with fate predictions, i.e. being eGFP+ at E18.5 and eGFP<sup>Bright</sup> at P0 and P6 (Fig. 5 and Supplementary Fig. 5). These findings not only identify previously undescribed markers of SSC fate specification and identity in the mouse germline, but also validated that the transcriptome signatures defined by the scRNA-seq database are reflected at the protein level.

**Regulatory networks associated with SSC fate specification.** Having defined different populations of prospermatogonia and initial postnatal spermatogonial subsets, we aimed to delve further into the pathways guiding the different fates. To start, cells from clusters 1, 5, and 7 of the complete dataset were extracted and analyzed with Monocle[48], which utilizes machine-learning algorithms to compare all single-cell transcriptomes in multidimensional space and orders cells in an unbiased manner along a path representing a developmental trajectory in theoretical time, known as pseudotime. We postulated that ordering transcriptomes along a developmental trajectory could reveal regulatory networks that dynamically change during SSC specification and establishment. Outcomes of trajectory analysis showed the correct ordering of developmental ages in pseudotime (Fig. 6a). Consistent with the clustering analysis performed above, germ cells from P3 and P6 overlapped along the ordered trajectory, indicating similarities among the population at P3 and P6 and supporting the conclusion of SSC establishment at P3 drawn from FCA (Fig. 2). Interestingly, germ cells from P0 and P3 also overlapped, which likely underscores the asynchronous nature of lineage maturation.

Utilizing the ordered trajectory, we next performed DEG analysis to identify changes as a function of pseudotime (gene list located in Supplementary Data 2). Four broad patterns of gene expression dynamics were identified (Fig. 6b) and analyzed further. Group 1 genes were upregulated early along the trajectory and declined shortly thereafter, representing a pattern consistent with SSC specification. Interestingly, a number of factors within the transforming growth factor β (TGFβ) and Wingless-Type MMTV Integration Site (Wnt) families were enriched in Group 1 (Fig. 6c). Group 2 represented a smaller set of genes similar to

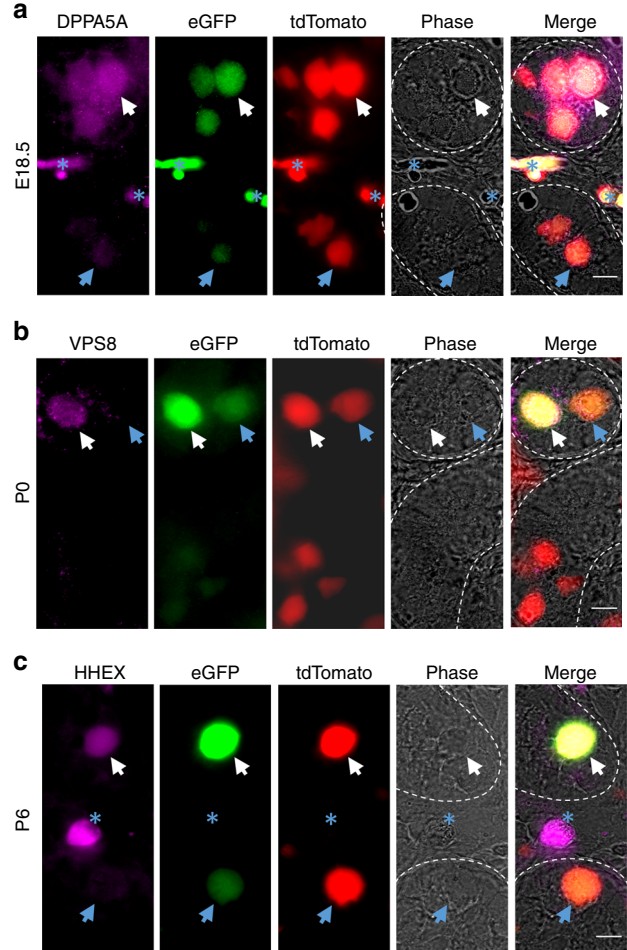

**Fig. 5** Validation of markers of SSC fate specification identified by scRNA-seq clustering. **a–c** Immunofluorescent staining for DPPA5A (**a**), VPS8 (**b**), and HHEX (**c**) proteins that were identified as differentially expressed at the transcript level in germ cell clusters 1, 5, and 7 of E16.5, P0, and P6 testes by scRNA-seq, respectively. Immunostaining is overlaid with ID4-eGFP and tdTomato fluorescence in germ cells. White arrows indicate germ cells that are DPPA5A+ and ID4-eGFP+ at E18.5, VPS8+ and ID4-eGFP<sup>Bright</sup> at P0, or HHEX+ and ID4-eGFP<sup>Bright</sup> at P6. Blue arrows indicate germ cells that have low to undetectable staining for the selected marker and ID4-eGFP. Blue asterisks denote in (**a**) vasculature in E18.5 testes that has autofluorescence, and in (**c**) an interstitial cell (i.e. tdTomato−) that is HHEX+. Seminiferous tubule borders are indicated by with white dotted lines. Scale bars, 10 µm. Images are representative of ≥3 cross-sections imaged from n = 2 biologically independent animals for each developmental age point

Group 1, but with more prolonged expression along the trajectory; these genes included a number of unique transcription factors and signaling proteins (Fig. 6d). Group 3 encompassed genes that peaked midway during the trajectory consistent with the transition from prospermatogonia to spermatogonia; these genes included a number of transcription factors, receptors, and metabolic regulators (Fig. 6e and Supplementary Fig. 6a). Consistent with the known migration of prospermatogonia from the seminiferous lumen to the basement membrane between P0 and P3, Group 3 genes were also enriched for cellular processes involved in cell movement, including focal adhesion, cell-to-cell adhesion, and cytoskeletal proteins. Group 4 genes increased in expression later along the trajectory consistent with involvement in the established SSC population. Also, Group 4 included a

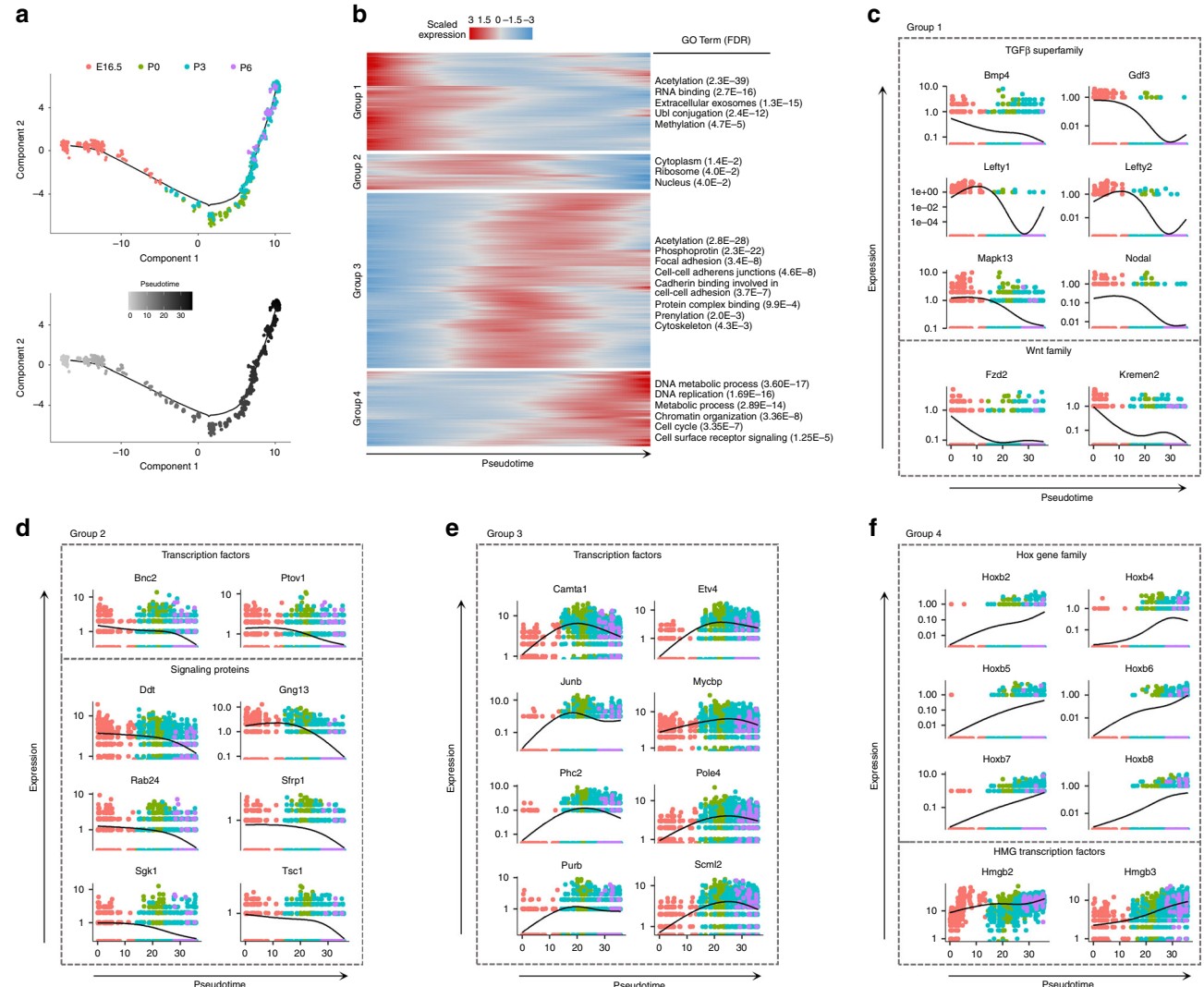

**Fig. 6** Transcriptional dynamics and regulatory networks within the SSC trajectory. **a** Trajectory analysis of SSC-enriched clusters from each developmental age ordered through pseudotime. **b** Heatmap representation of four broad gene expression patterns that vary as a function of pseudotime. Associated gene ontology (GO) terms and false discovery rates (FDR) are reported for each group. **c–f** Select protein families and pathways that are differentially expressed across pseudotime corresponding to gene groups in **b**. Dot coloring in **c–f** corresponds to developmental ages illustrated in **a**

number of genes involved in DNA replication and cell cycle progression, as well as several homeobox (Hox) and high mobility group (HMG) family transcription factors (Fig. 6f). A number of unique chromatin-associated transcription factor and receptor genes were also identified in Group 4 (Supplementary Fig. 6b).

**Stem cell capacity of fetal prospermatogonial subsets**. Collectively, the developmental kinetic map and lineage trajectory predictions from transcriptome profiling suggest that SSC fate is preprogrammed in a subset of prospermatogonia during late fetal development. To begin exploring this on a functional level, we performed transplantation analysis of defined prospermatogonial subsets isolated from testes of quadruple-transgenic mice at E18.5 (Fig. 7a). Germ cell transplantation has been the standard in the field of spermatogenesis for functional assessment of SSC capacity for over two decades[49–52]. In this assay, only cells endowed with regenerative capacity will give rise to colonies of spermatogenesis within recipient testes, thus providing unequivocal evidence of fulfilling the functional definition of a stem cell. Prospermatogonia (marked as tdTomato+ cells) were FACS isolated from testes of E18.5 fetuses based on a distinguishing feature of

being ID4-eGFP+ or ID4-eGFP− and transplanted into testes of adult germ cell-depleted recipient males. Two months later, colonies of donor-derived spermatogenesis were assessed via X-gal staining.

Two predictions of the outcomes could be made for this experimental paradigm: (1) if all prospermatogonia are pliable and SSC fate is stochastically determined, both ID4-eGFP+ and ID4-eGFP− prospermatogonial subsets would give rise to similar numbers of donor-derived spermatogenic colonies, or (2) if SSC fate is preprogrammed in a defined subset of prospermatogonia, donor-derived spermatogenic colonies would arise from either ID4-eGFP+ or ID4-eGFP− cells. Outcomes of the transplantation analyses revealed that only cells present in the ID4-eGFP+ fraction of prospermatogonia possessed the functional stem cell capacity to regenerate the spermatogenic lineage (Fig. 7b, c). While the ID4-eGFP+ cells generated $177.9 \pm 115.4$ colonies/$10^5$ cells transplanted ($n = 14$ recipient testes, three biologically independent cell suspensions, and 2366 total cells), no colonies were observed in testes transplanted with ID4-eGFP− cells ($n = 15$ recipient testes, three biologically independent cell suspensions, and 20,002 total cells). Taken together, these findings

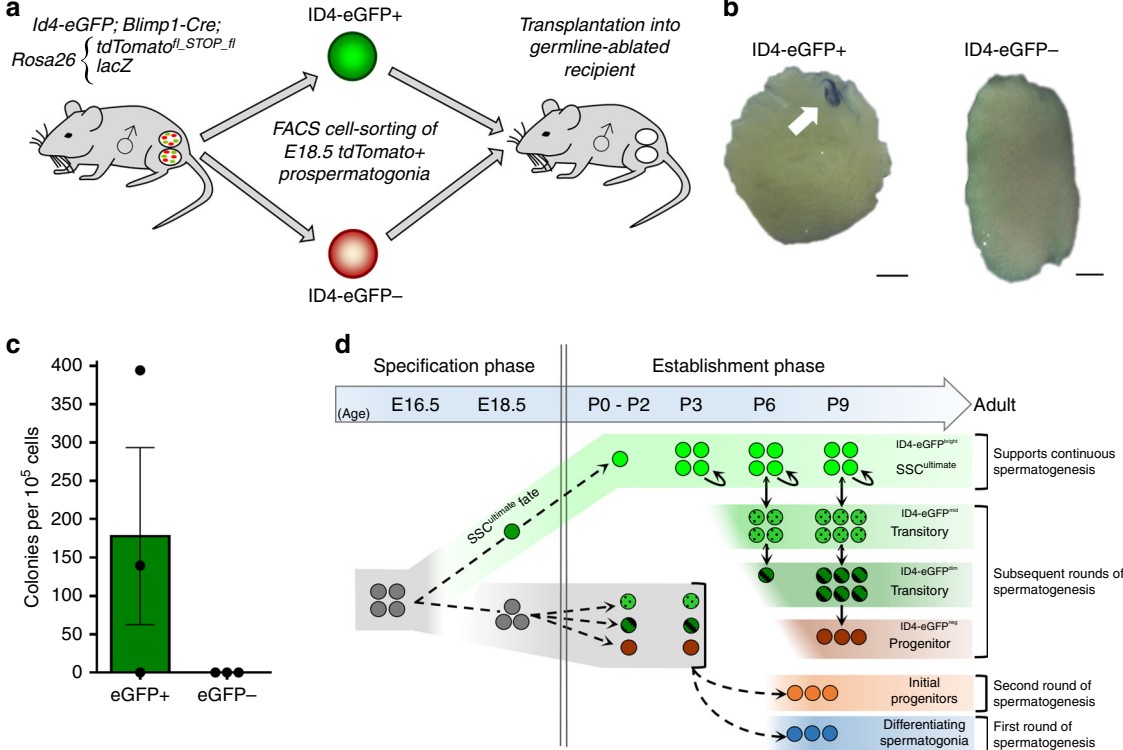

**Fig. 7** Functional assessment of SSC fate specification in subsets of fetal prospermatogonia. **a–c** Schematic of transplantation experimental approach **a**, representative images of recipient testes after X-gal staining to discern donor-derived spermatogenic colonies (**b**; scale bars, 1 mm), and stem cell quantification **c**. Data are presented as means with error bars representing SEM for n = 3 biologically independent donor prospermatogonial suspensions divided into either ID4-eGFP+ (2366 total cells) and ID4-eGFP− (20,002 total cells) and transplanted into 14 or 15 recipient testes, respectively. Source data are provided as a Source Data file. **d** Proposed model explaining the kinetic mechanisms that guide establishment of spermatogonial subsets, including the foundational SSC pool, during neonatal life from prospermatogonial precursors that were preprogrammed in late fetal development

support a model of SSC fate being preprogrammed in a defined subset of prospermatogonia during late fetal development.

## Discussion

The mechanisms and kinetics underpinning SSC population establishment during development have remained a black box. Therefore, we sought to explore when and how the foundational SSC pool forms. We found that specification of SSCs and establishment of the foundational pool are distinct events defined by a biphasic model (Fig. 7d). Outcomes from surveying ID4-eGFP transgene expression and scRNA-seq analyses revealed that expression of core SSC factors initiates as early as E16.5 in a subset of prospermatogonia during the specification phase. Furthermore, transplantation analyses provided compelling evidence indicating that an underlying program endows a subset of male germ cells with stem cell capacity by E18.5. Thus, contrary to our current understanding, postnatal spermatogonial fate may be predetermined within a subset of the heterogeneous prospermatogonial population during fetal development. In the subsequent establishment phase, preprogrammed prospermatogonia reenter mitosis from P0 to P3 and self-renew to build the foundational SSC pool to a theoretical limit (~12,500 per testis in the mouse), as defined by expansion and then maintenance of the ID4-eGFP$^{Bright}$ population. Subsequent layers of transitioning spermatogonia, identified by decreasing ID4-eGFP expression, then arise from the established SSC population between P3 and P9. These observations suggest a shift from symmetrical to asymmetrical division during mammalian male germline development similar to invertebrate models[53,54]. However, further studies are necessary to visualize in vivo asymmetric cell division

by the established SSC population before concrete conclusions can be drawn.

In addition to defining the dynamics of SSC pool establishment during the neonatal period, the kinetic mapping and fate trajectory predictions from scRNA-seq were able to infer subsets of prospermatogonia at P0 that give rise to other postnatal spermatogonial states. The subset giving rise to initial differentiating spermatogonia directly (delineated as ID4-eGFP−) was found to be present at birth but did not expand in number until P3–8, after SSC pool establishment. In corroboration, nearly all differentiating spermatogonia at P3 (delineated as c-KIT+) are ID4-eGFP−. Interestingly, a subset of prospermatogonia that seemingly gives rise to initial progenitor spermatogonia directly (delineated as ID4-eGFP$^{Mid}$) is present shortly after birth but also does not expand in number until after P3. Because most of these cells are c-KIT− at P3 but ~50% of the ID-eGFP$^{Mid}$ population is c-KIT+ at P6[21], the possibility that this population may contribute to a unique second round of spermatogenesis is a tempting postulation. Another intriguing speculation is that both of these non-SSC-fated postnatal trajectories derive from the ID4-eGFP− fetal prospermatogonial subset, which is supported by an inability to engraft in recipient testes after transplantation. However, future experimentation using complimentary approaches such as lineage tracing are necessary to functionally test these possibilities.

Beyond mapping the kinetics of spermatogonial lineage formation, we also report that prospermatogonial precursors and descendent SSCs spatially cluster as nests during development. Furthermore, nests are composed of germ cells of common fate, either as SSCs (ID4-eGFP$^{Bright}$) or cells transitioning to a progenitor state (ID4-eGFP$^{Mid/Dim}$). Interestingly, previous studies

showed that mammalian prospermatogonia form intercellular connections in fetal and neonatal gonads[11,37], and clonal analysis illustrated that connected prospermatogonia are of common genetic origin[36,55,56], suggesting that interconnected germ cells adopt a common fate. Similar intercellular connections form among female germ cells in the fetal ovary. However, contrary to the male germline, fate determination between primary oocytes and apoptotic follicles occurs within each interconnected group of germ cells[57]. Among prospermatogonia, syncytial connections are known to be progressively lost during late fetal development[56], with some newborn prospermatogonia retaining intercellular bridges[58]. Based on observations made in the current study, syncytial interconnection among prospermatogonia of common fate does not appear to align with nest formation or breakdown. Therefore, the function of cytoplasmic connections among prospermatogonia in the male gonad remains unclear. Cytoplasmic connections between cells are known to drive synchronous cell cycle progression[57], and expression of cell cycle regulators is altered in cultures of spermatogonia derived from *Tex14* null mice that lack intercellular bridges[59]. Thus, the synchronous mitotic arrest and reactivation of male germ cells during development may require cross-communication via cytoplasmic interconnectedness. Importantly, however, studies from TEX14 knockout mice reported no defects in the spermatogonial population[60]. Interestingly, spermatogenesis progresses up to meiosis in *Tex14* null mice, thus implying an intact spermatogonial compartment and normal SSC pool formation. However, the fact that the spermatogenic lineage does not turnover in *Tex14* null mice due to a block in the differentiation process makes assessing regeneration of the spermatogonial population difficult under homeostatic conditions. Regardless, findings of the current study indicate that while the dynamics of nest breakdown differ among the SSC (ID4-eGFP^Bright) and transitory (ID4-eGFP^Mid/Dim) populations, the act of germ cell clustering and the size of nests are not indicative of cell fate. Rather, based on our scRNA-seq analysis, we postulate that the surrounding soma creates microenvironments (e.g. niches) that influence SSC specification and establishment of the foundational pool as nests.

Trajectory analysis of the SSC lineage through pseudotime provided further clues about the localized signals that may be emanating from the soma to influence SSC specification. In particular, our analyses identified components of the TGFβ and Wnt signaling pathways during specification of SSCs, suggesting that extrinsic stimuli drive initial fate determination. Likewise, sets of receptors both previously characterized (*Ret*[61], *Mcam*[62], *Bmpr1a*[63], etc.) and unexplored were upregulated at various stages along the SSC trajectory. In support of the concept that somatic cells participate in nest formation, previous studies identified patches of RA production along seminiferous cords that are associated with germ cell differentiation in the testis at P2[64]. As RA is predominantly produced by Sertoli cells in neonatal testes[43,65], this finding suggests heterogeneity exists within the soma during early neonatal development. Additionally, differential expression of galectin 1 (*Lgals1*) among fetal Sertoli cells at E18.5 suggests that somatic heterogeneity extends to prenatal development as well[66]. Somatic patterning has been attributed to the onset of asynchronous differentiation along the length of a seminiferous tubule[43]. However, the grouping of germ cells and surrounding somatic cells into nests may reflect paracrine communication to create stem cell-niche units throughout development, as proposed previously[1].

Outcomes of our scRNA-seq studies also revealed broad patterns of gene expression that coincide with phases of SSC pool formation. Aligning these patterns along a developmental continuum provides valuable insight into networks regulating SSC functions

and illuminates new transcriptional players involved in SSC pool formation. For example, after birth numerous transcription factors, receptors, and metabolic regulators are upregulated during SSC pool establishment (Fig. 6, Group 3 genes). In particular, *Etv4*, a member of the E26 transformation-specific transcription factor family that shares significant relation to the SSC regulator ETV5[67]. Interestingly, an ETV4 consensus-binding sequence lies upstream of the coding sequence for *T* (*Brachyury*)[68], which is a known regulator of SSC maintenance. Furthermore, based on our scRNA-seq profiling, *T* expression is upregulated along the SSC trajectory following *Etv4*. Thus, temporal patterns in gene expression reveal potential regulatory systems that guide building of the SSC pool.

A striking observation from scRNA-seq profiling is the presence of transcriptome continuums at all developmental ages we examined. As opposed to forming populations that segregate based on transcriptomic signature and accompanying fate—for example, when visualized by tSNE or UMAP—SSCs and their precursors arranged along a continuum opposite germ cells destined for a differentiating fate. Interestingly, transcriptional gradients have been observed in neural[69], epidermal[70], and hematopoetic[71] stem cell lineages under steady-state conditions. Our scRNA-seq analysis suggests that relative position along a continuum during development drives stem cell specification. Given that the *Id4-eGfp* transgene faithfully represents endogenous *Id4* expression[18], sampling based on eGFP intensity through developmental time provided further in vivo evidence to support continuums observed from the scRNA-seq studies. Finally, transplantation analyses solidified that expression of core SSC regulators is indeed indicative of fate.

Collectively, the findings presented in the current study demonstrate that establishment of the foundational SSC pool is triggered from a subset of prospermatogonia at P0 and completes by P3 (Fig. 7d). What is the mechanism underpinning this process? Two concepts can be logically envisioned: (1) preprogramming within a subset of fetal prospermatogonia that locks-in SSC fate or the capacity to attain SSC fate in neonatal development versus adopting a differentiating fate or (2) equal propensity of all prospermatogonia to become SSCs or differentiating spermatogonia during neonatal development with stochasticity driving the different fate trajectories. The observation that defined subsets of prospermatogonia can generate colonies of spermatogenesis following transplantation into adult testes suggests that the propensity for SSC fate is already in place within a portion of the population at E18.5. Although this experimental approach creates an anomalous situation because prospermatogonia are not present in adult testes under normal circumstances, the capacity to engraft after transplant could be viewed as the ultimate functional test of spermatogenic stem cell capacity. While studies by McLean et al. indicated that donor germ cells prior to P4 are unable to colonize adult recipient testes[72], studies of Kubota et al. demonstrated that a subset of prospermatogonia at P0–1 can engraft in adult recipient testes and generate colonies of spermatogenesis[73], consistent with findings in the present study. Importantly, approximately ten-fold more ID4-eGFP− prospermatogonia than ID4-eGFP+ prospermatogonia isolated from E18.5 testes were transplanted and donor-derived colonies from the ID4-eGFP− subpopulation were not observed. Although seemingly paradoxical, we also observed that the size of ID4-eGFP^Bright nests are relatively constant, but the total number of nests increase from P0 to P2, suggesting that the subset of prospermatogonia serving as the seed population of the foundational SSC pool forms during neonatal development. Based on these collective observations, we propose a conceptual model in which the capacity to attain SSC fate during neonatal development is not stochastic but rather preprogrammed in a subset of prospermatogonia during fetal development with a different

subset of fetal prospermatogonia programmed to become the first differentiating spermatogonia in neonatal development. This model is consistent with findings of Kluin and de Rooij that observed subsets of prospermatogonia in fetal testes possessing nuclear morphologies that resembled either undifferentiated or differentiating postnatal spermatogonial subtypes[11]. While intriguing, validation of this preprogramming concept will require future experimentation using complimentary approaches, such as lineage tracing and ablation to link trajectory predictions of all fetal prospermatogonial subsets to functional fates in postnatal life.

## Methods

**Animals**. All procedures for the ethical and humane use of animals in the present study were approved by the Washington State University Animal Care and Use Committee. To generate a multi-transgenic reporter model for investigating germ cell subsets, separate breeder lines were generated before performing a final cross. First, *Blimp1(Prdm1)-Cre* (Jackson Laboratories, stock no. 008827), *Id4-eGfp* (generated previously[18]), and *Rosa26-LacZ* (Jackson Laboratories, stock no. 002073) transgenic mice were bred to generate triple-transgenic founders (*Blimp1-Cre^Tg; Id4-eGFP^Tg; Rosa26^LacZ/LacZ*). Second, *Rosa26-tdTomato^flox_STOP_flox* (Jackson Laboratories, stock no. 007909) and *Id4-eGfp* transgenic mice were crossed to generate double-transgenic founders (*Id4-eGFP^Tg; Rosa26^tdTomato-fl_STOP_fl/tdTomato-fl_STOP_fl*). Finally, quadruple-transgenic males for analysis at E16.5-P9 were generated from crossing *Blimp1-Cre^Tg;Id4-eGFP^Tg;Rosa26^LacZ/LacZ* males with *Id4-eGFP^Tg;Rosa26^tdTomato-fl_STOP_fl/tdTomato-fl_STOP_fl* females to generate *Blimp1-Cre^Tg;Id4-eGFP^Tg;Rosa26^tdTomato_fl/LacZ* males. Recombination was estimated at >98% based on the percentage of neonatal eGFP+ spermatogonia that were also tdTomato+. Notably, *Id4-eGfp* transgene expression levels are unaffected by zygosity. Adult male mice for analysis (>P56) possessed the *Id4-eGfp* transgene only.

**Flow cytometric analysis**. Single cell suspensions were generated from isolated embryonic and neonatal testes by trypsin/EDTA digestion. Briefly, detunicated testes were incubated in a solution of 0.25% trypsin/EDTA (Thermo Fisher Scientific) and 2 mg/mL deoxyribonuclease I (Sigma-Aldrich, Inc.) for 10 min at 37 °C with gentle agitation. Trypsin digest was quenched with 10% fetal bovine serum (FBS) before cells were washed and resuspended in a solution of 1% FBS, 10 mM Hepes, 1 mM sodium pyruvate, 1 mg/mL glucose, 100 units/mL penicillin, and 100 µg/mL streptomycin in PBS (Thermo Fisher Scientific or Sigma-Aldrich, Inc.). For adult mice, testes were first incubated in 1 mg/mL collagenase type IV (Thermo Fisher Scientific) at 37 °C for 10 min with gentle agitation to disperse seminiferous tubules, rinsed three times with HBSS on ice to remove interstitial cells, and single cell suspensions generated by digestion with trypsin/EDTA solution. Because the *Id4-eGfp* transgene is expressed by pachytene spermatocytes, as well as SSCs, testis suspensions from adult *Id4-eGFP* males were incubated with an antibody recognizing the undifferentiated spermatogonial cell surface marker CDH1 (E-Cadherin) [74] (Biolegend, clone DECMA-1, catalog no. 147307) at a dilution of 1:200 for 30 min on ice and then gently washed three times before analysis. For KIT staining, cell suspensions were incubated with a fluorophore-conjugated antibody recognizing KIT (Abcam, clone 2B8, catalog no. ab25495) at a dilution of 1:100 for 30 min on ice and then gently washed three times before analysis. Cell suspensions were analyzed with an Attune Nxt Flow Cytometer (Thermo Fisher Scientific; software v2.7.873) and data processed with Kaluza Analysis Software (Beckman Coulter, Inc.; software v1.5a).

Background fluorescence was determined from unstained control testis suspensions of equivalent age. Single-stained fluorescent controls were used for calculating compensation. A representative gating strategy for isolating all tdTomato+ germ cells is presented in Supplementary Fig. 2d. For each sample, eGFP fluorescent signal was equally divided into thirds to establish Bright, Mid, and Dim designations, consistent with our previous studies[20] and illustrated in Supplementary Fig. 2a, b. It is important to note from analysis in Fig. 2, adult eGFP− spermatogonia constitute undifferentiated spermatogonia gated from CDH1 expression, while embryonic and neonatal eGFP− germ cells represent undifferentiated spermatogonia and the remaining eGFP− germline, which includes differentiating spermatogonia. Thus, the two eGFP− populations consist of different cellular populations and therefore, adult eGFP− cells were omitted from the analysis.

**Cell cycle staining**. Single cell suspensions were fixed in 4% paraformaldehyde/PBS for 6 min at 37 °C, permeabilized in 90% methanol for 30 min on ice, and finally incubated with 50 µg/mL RNase A (Thermo Fisher Scientific) and FxCycle Violet or Far Red (Thermo Fisher Scientific) according to the manufacturer's protocol. Adult cell suspensions were first incubated with the CDH1 antibody, then fixed, permeabilized, and stained with FxCycle dye. All cell cycle staining was then analyzed with an Attune Nxt Flow Cytometer (Thermo Fisher Scientific; software v2.7.873) and data processed with Kaluza Analysis Software (Beckman Coulter, Inc.; software v1.5a).

**Confocal imaging of whole testes and nest quantification**. After removal of the tunica albuginea, isolated testes were fixed in 4% paraformaldehyde/PBS for 2 h at 4 °C, rinsed three times in PBS at room temperature for a total of 4 h, and then cleared in ScaleS4(0) solution[75]. ScaleS4(0) solution was replaced twice daily until the tissue became transparent—typically after 2–3 days. Cleared individual testes were imaged in ScaleS4(0) solution in a glass-bottom microwell dish (MatTek Corporation) by a TCS SP8 X confocal laser scanning microscope (Leica Microsystems; LasX v3.3.0) using a ×10 objective. Optical slices were taken at 5 µm increments through the entire depth of the testis (E18.5–P3) or until fluorescent signal began to dissipate (P5–P9), and Z-compensation was applied to maintain uniform fluorescence. Adjacent Z-stacks were overlapped by >15% for accurate image stitching. Confocal imaging could only sufficiently identify ID4-eGFP^Bright and ID4-eGFP^Mid populations without saturating fluorescence.

Aggregate eGFP images were analyzed in ImageJ (v1.51t) using the '3D ROI Manager' plugin to determine distances separating each detectable object along with object volume and eGFP intensity. Using R software (v3.4.4), object distances were then used to perform single-linkage hierarchical clustering (hclust function) with a cutoff distance of 10 µm. Object volume was used to estimate cell number, and average eGFP intensity within a nest determined the eGFP classification. Due to imaging depth limitations, approximately the top half of P5, P7, and P9 testes were imaged; therefore, total nest number was estimated based on the number of cells from each eGFP+ subtype captured through imaging versus the total number of cells for each eGFP+ subtype quantified by FCA in Figs. 1 and 2 (denoted with asterisks in Fig. 3d).

**Single-cell RNA-sequencing analysis**. Single cell suspensions of E16.5, P0, P3, or P6 quadruple-transgenic testes were prepared and germ cells isolated using FACS with an SH800 machine (Sony Biotechnology) gating for tdTomato+ cells. Live cells were loaded into a Chromium Controller (10X Genomics, Inc.) and single-cell cDNA libraries were generated using v2 chemistry according to the manufacturer's protocol (10X Genomics, Inc.). Libraries were pooled at proportions netting equal read depth and sequenced in a single lane on an Illumina HiSeq 4000 (Genomics and Cell Characterization Core Facility, University of Oregon). Raw base call files were demultiplexed using the 10X Genomics Cell Ranger pipeline (v2.1.0) and aligned to the mouse mm10 transcriptome.

A total of eight transcriptomes ($n = 3$ for E16.5, $n = 2$ for P0, $n = 2$ for P3, and $n = 1$ for P6) were merged in R using the Seurat package[39] (v2.3.2). Low-quality cell transcriptomes and doublets were assessed within each library and excluded using the following criteria: >1300 genes for E16.5 #1; >1900 genes and <80,000 UMI counts for E16.5 #2; >1200 genes for P0 #1; >1000 genes and <30,000 UMI counts for P0 #2; >1500 genes and<70,000 UMI counts for P3 #1; <35,000 UMI counts for P3 #2; <40,000 UMI counts for P6. Filtering based on percent mitochondrial genes detected was conservatively set at <25% for all libraries based on visual inspection of overall distribution and the variability of mitochondrial gene content that occurs based on cell type, proliferation status, and developmental state. Germ cells were isolated from a small number of contaminating cells in the dataset based on *Dazl* and *Ddx4* transcript abundance before the entire dataset was normalized, scaled ("vars.to.regress" included number of genes, number of UMIs, percent mitochondrial transcripts, and age replicate), and dimensionally reduced using Seurat. Variable genes were identified based on average expression and dispersion, followed by binning genes based on average expression, and calculating z-scores for each bin (Seurat "FindVariableGenes" function). Identified variable genes were used to perform principal component analysis and 31 significant ($p < 0.01$) principal components were selected for clustering ("resolution" set to 0.6) and tSNE graphing. The SimplifyStats package (v1.0.1) was used to calculate cluster and age distribution. Modified multivariate Pearson's RV correlations[24] were calculated for each set of age replicates using the MatrixCorrelation package (v0.9.2) in R. The following correlations suggest negligible batch effects between replicate libraries: E16.5 #1 and E16.5 #2 = 0.96; E16.5 #2 and E16.5 #3 = 0.99; E16.5 #1 and E16.5 #3 = 0.96; P0 #1 and P0 #2 = 0.99; and P3 #1 and P3 #2 = 0.99.

Clusters 1, 5, and 7 were isolated from the complete dataset and imported into the Monocle 2 package (v2.6.4) of R[76] for trajectory analysis using the recommended "dpFeature" approach. No additional filtering of transcriptomes was performed from above and 11 dimensions were selected for dimensional reduction. Pseudotime DEG analysis was performed using Monocle 2 with a q-value cutoff of <0.01. Gene Ontology was performed using PANTHER (v13.1) or DAVID (v6.8) software using a list of all detected genes from the entire scRNA-seq library as background.

**Immunostaining of testis cross-sections**. Testes isolated from E18.5, P0, or P6 quadruple-transgenic mice were fixed with 4% paraformaldehyde/PBS for 2 h at 4 °C, rinsed three times in PBS for 30 min at 4 °C, and then immersed in 15% sucrose/PBS solution overnight at 4 °C. Testes were then incubated in a solution of 7.5% gelatin, 15% sucrose, and 1X PBS for 1 h at 37 °C before snap-frozen in a bath of isopentane at −55–65 °C. Frozen sections at 10 µm thickness were melted onto glass slides and incubated for >2 h at room temperature for adherence. Samples

were permeabilized in a solution of 0.1% Triton X-100 before incubating in blocking buffer (0.1% fish skin gelatin, 0.5% Triton X-100, 0.1% Tween-20, and 10% normal goat or donkey serum in PBS) for 1 h at room temperature. Slides were then incubated in primary antibody diluted in blocking buffer overnight at 4 °C. Antibody conditions and information are as follows: DPPA5A (ThermoFisher, catalog no. PA5-48042, lot UB2715024A, diluted 1:20–1:40), VPS8 (ThermoFisher, catalog no. PA5-57855, lot UB2715146A, diluted 1:50–1:100), HHEX (Thermo-Fisher, catalog no. PA5-19625, lot UB2714724, diluted 1:50–1:100). After primary antibody incubation, slides were washed thrice in 0.2% Tween-20/PBS before incubating with fluorescent secondary antibodies (Thermo Fisher Scientific) for 2 h at room temperature. Finally, slides were washed in PBS three times, mounted (Vector Laboratories, catalog no. H-1500), and imaged with a DMi8 inverted fluorescent microscope (Leica Microsystems; LasX v3.3.0) using a ×63 objective. Digital images were processed using ImageJ (v1.51t).

**Transplantation of prospermatogonial subsets**. To assess the capacity of pros-permatogonial subsets to regenerate the spermatogenic lineage and therefore fulfill the functional definition of an SSC, transplantation analyses were conducted[20]. Briefly, donor cell suspensions were prepared from the testes of individual E18.5 quadruple-transgenic fetuses. All tdTomato+ prospermatogonia were isolated and separated into ID4-eGFP+ and ID4-eGFP− fractions via FACS. Isolated popula-tions were washed by centrifugation and resuspended in mouse serum-free med-ium. The ID4-eGFP+ and ID4-eGFP− fractions were then transplanted into the testes of busulfan-treated recipients via microinjection. For each recipient, one testis received ID4-eGFP+ cells and the contralateral testis received ID4-eGFP− cells. Two months after transplantation, donor-derived spermatogenic colonies were assessed by X-gal staining and counted under a dissection microscope.

**Statistical analysis**. Cell number, eGFP distribution, cell cycle, and nest quanti-fications were analyzed by Student's t-test or one-way ANOVA using Prism soft-ware (GraphPad Software; v6.07). All quantitative data are presented as means with error bars representing standard error of the mean (SEM).

**Reporting summary**. Further information on research design is available in the Nature Research Reporting Summary linked to this article.

## Data availability
Source data generated and analyzed for FCA (Figs. 1 and 2), nest quantification (Fig. 3), and transplantation (Fig. 7) experiments in this study are included as a Source Data file. Transcriptome (scRNA-seq) data generated and analyzed during the present study are available from the GEO database (accession number GSE124904).

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

## Acknowledgements

The authors wish to thank Dr. Aileen R. Helsel for assistance with derivation of transgenic mouse lines and tubule imaging, as well as the other members of the Oatley lab for their insight and expertise. This research used resources from the Center for Institutional Research Computing at Washington State University Grant HD061665 awarded to J.M.O. from the NICHD.

## Author contributions

Conceptualization and methodology: N.C.L. and J.M.O.; formal analysis: N.C.L.; investigation: N.C.L. and M.J.O.; writing—original draft; N.C.L. and J.M.O.; writing—review and editing, N.C.L., M.J.O., and J.M.O.; funding acquisition: J.M.O.

## Additional information

**Competing interests:** The authors declare no competing interests.

