## [Peer Review File · Nature Communications]

Reviewers' comments:

Reviewer #1 (Remarks to the Author):

This study by Law et al. characterizes the development of spermatogonial stem cells (SSCs) from fetal germ cells (prospermatogonia/gonocytes) in the developing mouse. SSCs are critical for maintaining spermatogenesis and fertility in males but mechanisms involved in SSC development from prospermatogonia poorly appreciated. It is known that prospermatogonia directly generate both differentiating spermatogonia (for first wave of spermatogenesis) and SSCs in neonatal testis and different models are proposed to account for balanced adoption of these different fates. To define mechanisms of SSC development, the authors characterize germ cell heterogeneity in fetal and neonatal testis by single cell RNA-Seq and analyze development of the Id4-expressing spermatogonial population, which contains SSCs. The approach is well designed and takes advantage of transgenic mice in which different fluorescent reporters mark the germ lineage and ID4-positive populations. The single cell analysis is informative and provides a good resource for the field. Observations concerning germ cell "nests" in neonatal testis are intriguing although somewhat preliminary. From this data the authors conclude that SSCs develop from a predetermined prospermatogonia subset defined by high levels of expression of SSC regulators and pluripotency genes. Also, that SSCs divide symmetrically after formation then asymmetrically at later stages to generate committed progenitor populations while maintaining SSCs. This point is based on dynamics of Id4 expression during development. In general, data are of high quality and the manuscript addresses important questions. However, presented data are not sufficient to support conclusions of the authors. Additional evidence for the proposed model and clarification of existing datasets need to be provided as listed below:

1. One limitation of this study is that it is descriptive in nature and does not confirm predictions from analysis of single cell RNA-Seq datasets with functional data. To support the claim that prospermatogonia expressing high levels of pluripotency and SSC genes at E16.5 form the SSC population in neonates, in vivo lineage-tracing studies should be performed. It is concluded that the data supports a model by which SSC fate is pre-programmed in prospermatogonia according to gradients in expression of stem and progenitor-associated genes. If the SSC marker-low population is pulse-labelled prior to birth, are lineage-marked cells lost through differentiation in the postnatal testis? If SSC-marker low and high cells of fetal/neonatal testis are transplanted, are stable colonies only generated from the SSC-marker high population? Use of algorithms to predict developmental states from single cell RNA-Seq data can be informative. However, as expression of SSC-associated genes may be dynamic, it is important to provide functional data to support the proposed model.

2. From dynamics of Id4 reporter expression in the newly formed spermatogonial pool, it is concluded that ID4Bright SSCs divide symmetrically after birth to expand the SSC pool and then switch to asymmetric division to generate ID4Mid progenitors while maintaining the ID4Bright SSC population. This is highlighted as being the first in vivo evidence for asymmetric cell division by mammalian SSCs (page 7). Given the potential importance of this observation, direct visualization of asymmetrically dividing ID4Bright cells to generate ID4Bright and ID4Mid cells should be provided rather than inferring this mode of division from analysis of bulk populations. At the timepoints of asymmetric division, daughter cells that have just been generated but have not migrated apart should have differential levels of Id4 expression. Lineage-tracing approaches may also be informative in this regard. Alternative explanations for the data can be proposed. Id4 expression is induced by niche growth factor GDNF (Oatley et al., BOR 2011) and levels of GDNF production are high in neonatal testis and downregulated during the first postnatal week (Meng et al., Science 2000). Many ID4Bright cells are therefore found in neonatal testis when GDNF is induced but ID4Mid cells dominate when GDNF levels are downregulated during postnatal development. Variation in Id4 expression therefore simply follows changes in GDNF levels during development and does not indicate switches between symmetric and asymmetric division of SSCs. Can the authors exclude the possibility that many ID4Bright cells divide symmetrically to generate

ID4Mid cells while a minority of ID4Bright persist?

3. In relation to point #2, the authors base their model on the assertion that ID4Bright cells are SSCs, ID4Mid spermatogonia are progenitors and ID4Dim cells are at later stages of differentiation. However, while transplantation studies of ID4Bright and ID4Dim cells show enrichment of stem cell activity within the ID4Bright population, functional potential of the ID4Mid population has not been previously defined (Helsel et al., Development 2017). Is the stem cell capacity of ID4Bright and ID4Mid populations different? Further, given that most spermatogonia express Id4 to varying extents during the first postnatal week but an expanding population of differentiating spermatogonia should also be present, are any of the ID4Bright, ID4Mid or ID4Dim populations positive for differentiation marker c-KIT? The authors refer to ID4Bright cells as "ultimate" SSCs but this term is not in general use in the field and could be clarified further.

4. Observed clustering of spermatogonia in the developing testis by 3D confocal imaging of the whole testis is very interesting and worth reporting (Figure 3). However, from the images shown, it is difficult to appreciate this phenomenon and higher magnification examples of the ID4Bright and ID4Mid nests should be provided. In relation to this point, are the cells in the nests in syncytia or present as clumps of isolated cells? Assuming ID4Bright cells represent SSCs while ID4Mid cells are progenitors, the ID4Bright population might be present as clumps of single cells while ID4Mid cells are in syncytia. Given the limitations of visualizing potential nests/groupings of ID4Dim prospermatogonia at fetal stages, the authors could use the Blimp1 lineage-marking model to study these timepoints. For instance, at E16.5 when prospermatogonia are proliferating, groupings of cells may be apparent and link localized cell proliferation with nest formation (page 8).

5. From tSNE plots of single cell RNA-Seq data, it is concluded that there are opposing gradients in expression of stem and progenitor genes in germ cells at fetal and postnatal stages of development (Fig. 4 and supplementary Fig. 4). However, from the data shown this is not always evident. For instance, while gradients in SSC genes *Lhx1*, *Etv5* and *Id4* are apparent at E16.5, P0 and P6, they appear uniformly expressed in germ cells at P3. *Ret* also seems uniformly expressed in P3 germ cells while at E16.5 appears expressed at low levels in few germ cells (Supplementary Fig. 4). Given that *Gfra1* is the co-receptor for GDNF with *Ret* and *Gfra1* is associated with SSCs, the authors should show expression of *Gfra1* at all timepoints by tSNE plot. Gradients in expression of progenitor and differentiation genes are also not always evident at all timepoints. *Sohlh1* appears uniformly expressed at P0 and P3 while *Stra8* expression peaks in a population toward the middle of the tSNE plot at p6 (Fig. 4). The authors should confirm whether these markers are heterogeneously expressed as proposed at these timepoints by other analyses or other means of data presentation. For some of these genes, it should be confirmed that protein levels follow changes in levels of the mRNAs as the two may not always correlate. Additionally, the group of Yumiko Saga has previously reported that prospermatogonia are heterogeneous for expression of some SSC-linked genes in the fetal testis (Pui and Saga, Mech Dev 2017, 144:125-139). This paper should be quoted and results discussed in relation to this report.

6. Clustering analysis of single cell RNA-Seq data shown in Figure 4d seems counter-intuitive – why are cells in clusters 7, 8 and 10 split into multiple populations in the tSNE plot? For instance, a subset of cells from cluster 7 is found in the middle of the cluster 8 population and a fraction of cluster 10 cells are in the cluster 7 population. The authors should try other clustering methods, e.g. SC3 (Kiselev et al., Nature 2017). Changing parameters in the current algorithm may also improve clustering analysis. A dendrogram should be provided to help in visualizing cell clusters and discriminating genes. Key genes distinguishing clusters in Figure 4e should be displayed. Additionally, pre-filter thresholds used in the analysis (page 21 of Methods) are very different between developmental timepoints and even between replicates of the same timepoint. For instance, 70,000 UMI counts for P3 replicate #1 and 35,000 UMI counts for P3 replicate #2. Why is this? Appropriate quality control graphs for the datasets should be included in supplementary figures. The cut-off for low quality transcriptomes based on mitochondrial gene counts (25%) seems high – 10% is typically used. Only one animal is used for the P6 timepoint while 2-3

animals are used for other timepoints. This may limit robustness of the P6 dataset due to sample variation. Ideally, 2 biological replicates should be used for each timepoint.

7. One of the most interesting parts of the manuscript is pseudotime analysis of SSC clusters at different developmental timepoints (Fig. 5) as this might reveal regulators of SSC development. This component could be explored/discussed further. Can authors confirm by e.g. immunostaining, that some of the genes of interest are dynamically expressed during conversion of prospermatogonia to SSCs? Is the pattern of Hox gene expression in developing SSCs expected? It is mentioned that germ cells from P3 and P6 overlap in the predicted development timeline, suggesting that SSCs are defined by P3. However, there also appears large overlap between cells at P0 and P3 (Fig. 5a) so are SSCs be established earlier?

8. As a minor point - are the authors confident that all germ cells are lineage-marked by the Blimp1-Cre transgene in their model system? This may influence the single cell results and interpretation.

Reviewer #2 (Remarks to the Author):

This study investigated the transcriptome dynamic of mouse spermatogonial stem cell (SSC) formation. Using scRNA-seq coupled with other methods (e.g., 3D imaging), they found SSC were specified from a subset of precursor prospermatogonia during fetal development, suggesting a novel model regarding to stem cell fate determination. Specific comments:

(1) Fig. 4h: Sox2 is classified as pluripotency marker while Sox3 is classified as progenitor marker. However, Sox2 and Sox3 are functional similar. For example, a study has shown that "...ectopic expression of SOX2 in the testes functionally rescues the spermatogenic defect of Sox3 null mice." (<https://www.ncbi.nlm.nih.gov/pubmed/28515211>)

Interestingly, the Sox2 and Sox3 seem to be negatively correlated based on Fig. 4h.

(2) What is the definition of percentage of expression (what is the cutoff to define expressed genes) in Fig. 4h?

(3) Figure 4: When pooling different ages/mice into one analysis, do you investigate possible batch effects?

(4) For GO and Pathway analysis, what are the background genes (control genes)? Do you use all the genes or expressed genes as background?

(5) Page 11: For the 10 clusters identified, several marker genes can clearly characterize the function of each cluster. But how do you choose marker genes? For example, in the manuscript (Fig. 4h), there are three pluripotency markers (Dppa3, Nanog and Sox2). However, there are many other well-known pluripotency markers, such as Oct-4, Ssea1. Are they not expressed or not selected?

Reviewer #3 (Remarks to the Author):

This manuscript provides a description of mouse male germ development from embryonic day 16.5 onwards. While hitherto this development was described primarily at the morphological level or according to the expression of one or few genes, in this manuscript it has been done at the scRNA seq level and using using genetically engineered mice fluorescently tagged for germ cells (Blimp1)

and spermatogonial stem cells (SSCs) (Id4). Very interesting results were obtained about the development of the SSC population and the expression of various genes in SSCs and their differentiating descendants with age. This manuscript will provide a solid basis for further work in this field. The experimental design and the description of the results are excellent. I have but one important and 2 minor comments.

I do not know what definition of an asymmetrical division the authors use. For me it means that already at division the daughter cells of a stem cell are different and will develop in different directions. For this to establish, the authors would have to show differences in the daughter cells already at the time of their formation by division from their mother cell. Clearly, the authors do not provide any evidence for this to occur, like it was shown for example in *Drosophila* where the daughter cells remaining in contact with the hub cells are different from those pushed away from it, already during division. The daughter cells may just as well be similar to each other and develop differently later on, depending on whether or not the local environment is inductive towards differentiation. The authors should delete any reference about symmetric or asymmetric divisions. For example, in page 7 lines 7-10: It should read that gradually more of the SSC daughter cells were induced to differentiate.

Page 2, line 14: Sex determination takes place at about E12.5 and as the prespermatogonia become quiescent at E16.5, I would not call that "Shortly".

Page 8, lines 4-6: I do not understand the last part of this sentence, when cells are equally proliferative and clearly do proliferate why add "but not necessarily quiescent"? This should be clarified.

We are grateful for the constructive comments and criticisms of all three reviewers and have tried to address each with revisions to the text and inclusion of new data. Overall, we believe that by addressing the critiques, the merit of the manuscript has been significantly improved.

Response to Reviewer #1 Comments

Reviewer Comment 1: One limitation of this study is that it is descriptive in nature and does not confirm predictions from analysis of single cell RNA-Seq datasets with functional data. To support the claim that prospermatogonia expressing high levels of pluripotency and SSC genes at E16.5 form the SSC population in neonates, in vivo lineage-tracing studies should be performed. It is concluded that the data supports a model by which SSC fate is pre-programmed in prospermatogonia according to gradients in expression of stem and progenitor-associated genes. If the SSC marker-low population is pulse-labelled prior to birth, are lineage-marked cells lost through differentiation in the postnatal testis? If SSC-marker low and high cells of fetal/neonatal testis are transplanted, are stable colonies only generated from the SSC-marker high population? Use of algorithms to predict developmental states from single cell RNA-Seq data can be informative. However, as expression of SSC-associated genes may be dynamic, it is important to provide functional data to support the proposed model.

Author Response: We appreciate the reviewer's point and agree that functional data is a valuable compliment to the descriptive findings and the developmental trajectories predicted by the scRNA-seq data. To address this, we have performed transplantation experiments. A quadruple transgenic mouse line (Id4-eGfp; Blimp1-Cre; Rosa(26)^{tdTomato_fl_STOP_fl / LacZ}) was generated that directs Cre-mediated tdTomato expression in addition to constitutive LacZ expression. Germ cells (labeled as tdTomato+) from testes of E18.5 fetuses were isolated and divided into subsets defined by expression of ID4-eGFP (i.e. ID4-eGFP+ and ID4-eGFP- fractions) via FACS, then transplanted into germ-cell depleted recipients, and donor-derived spermatogenic colonies were assessed by LacZ staining two months later. Outcomes demonstrated that SSC-derived regenerative capacity within the germline is restricted to the ID4-eGFP+ fraction at E18.5, thus functionally confirming that SSC fate is preprogrammed within a defined subset of prospermatogonia during late fetal development. These new data have been incorporated into Figure 6 of the revised manuscript.

Although we agree that lineage tracing provides a complimentary functional analysis of cell fate, such analyses are not feasible at this time. This approach would require lineage tracing the ID4-eGFP- population in fetal development and to do so requires use of a gene(s) that is expressed solely in these cells which at present is unknown. For this reason, we chose to utilize transplantation as a functional measure of SSC fate specification.

Reviewer Comment 2: From dynamics of Id4 reporter expression in the newly formed spermatogonial pool, it is concluded that ID4Bright SSCs divide symmetrically after birth to expand the SSC pool and then switch to asymmetric division to generate ID4Mid progenitors while maintaining the ID4Bright SSC population. This is highlighted as being the first in vivo evidence for asymmetric cell division by mammalian SSCs (page 7). Given the potential importance of this observation, direct visualization of asymmetrically dividing ID4Bright cells to generate ID4Bright and ID4Mid cells should be provided rather than inferring this mode of division from analysis of bulk populations. At the timepoints of asymmetric division, daughter cells that have just been generated but have not migrated apart should have differential levels of Id4 expression. Lineage-tracing approaches may also be informative in this regard. Alternative explanations for the data can be proposed. Id4 expression is induced by niche growth factor GDNF (Oatley et al., BOR 2011) and levels of GDNF production are high in neonatal testis and downregulated during the first postnatal week (Meng et al., Science 2000). Many ID4Bright cells are therefore found in neonatal testis when GDNF is induced but ID4Mid cells dominate when GDNF levels are downregulated during postnatal development. Variation in Id4 expression therefore simply follows changes in GDNF levels during development and does not indicate switches between

symmetric and asymmetric division of SSCs. Can the authors exclude the possibility that many ID4Bright cells divide symmetrically to generate ID4Mid cells while a minority of ID4Bright persist?

Author Response: We agree and thank the reviewer for addressing the limitations in our observations of asymmetrical and symmetrical division. In eagerness to report the observations, we had over-interpreted the findings. While direct surveillance of each cell division modality *in vivo* would solidify our observations, there is a current lack of tools to visualize asymmetrical and/or symmetrical divisions under physiological conditions in mice, especially neonatal pups. Thus, we have revised the text to temper interpretation of the findings.

Reviewer Comment 3: In relation to point #2, the authors base their model on the assertion that ID4Bright cells are SSCs, ID4Mid spermatogonia are progenitors and ID4Dim cells are at later stages of differentiation. However, while transplantation studies of ID4Bright and ID4Dim cells show enrichment of stem cell activity within the ID4Bright population, functional potential of the ID4Mid population has not been previously defined (Helsel et al., Development 2017). Is the stem cell capacity of ID4Bright and ID4Mid populations different? Further, given that most spermatogonia express Id4 to varying extents during the first postnatal week but an expanding population of differentiating spermatogonia should also be present, are any of the ID4Bright, ID4Mid or ID4Dim populations positive for differentiation marker c-KIT? The authors refer to ID4Bright cells as “ultimate” SSCs but this term is not in general use in the field and could be clarified further.

Author Response: We thank the reviewer for acknowledging our lack of clearly explaining the functional differences among the ID4-Bright, -Mid, and -Dim populations. The appropriate explanations have now been incorporated into the Introduction section. Briefly, a previous publication from our lab (Lord et al., 2018) illustrated that Mid and Dim populations have the capacity to respond to retinoic acid (RA) *in vivo* and transition to a differentiating state (marked by attainment of c-Kit expression), while the ID4-eGFP^{Bright} spermatogonia do not. Thus, the Mid and Dim populations are functionally distinct from the Bright population. Transplantation of Bright and Dim populations only (Helsel et al., 2017) was due to the technical limitations of obtaining pure populations via FACS cell-sorting. Finally, the SSC^{Ultimate} nomenclature has only recently been suggested, but has been adopted by long-standing experts in the field (see deRoos, 2018).

Reviewer Comment 4: Observed clustering of spermatogonia in the developing testis by 3D confocal imaging of the whole testis is very interesting and worth reporting (Figure 3). However, from the images shown, it is difficult to appreciate this phenomenon and higher magnification examples of the ID4Bright and ID4Mid nests should be provided. In relation to this point, are the cells in the nests in syncytia or present as clumps of isolated cells? Assuming ID4Bright cells represent SSCs while ID4Mid cells are progenitors, the ID4Bright population might be present as clumps of single cells while ID4Mid cells are in syncytia. Given the limitations of visualizing potential nests/groupings of ID4Dim prospermatogonia at fetal stages, the authors could use the Blimp1 lineage-marking model to study these timepoints. For instance, at E16.5 when prospermatogonia are proliferating, groupings of cells may be apparent and link localized cell proliferation with nest formation (page 8).

Author Response: We agree that the nest phenomenon is an exciting observation for the field. The use of Blimp1-Cre-driven tdTomato expression to explore cytoplasmic connections is a great idea, however, we intentionally approached the exploration of syncytia with restraint. There are several studies that have reported intercellular connections among clonally-derived PGCs and prospermatogonia (Mork et al., 2012, Mech Dev; Niedenberger et al., 2018, Development; Lei and Spradling, 2013, Development); these studies are now referenced in the discussion. However, in mouse knockout models that lack intercellular connections among germ cells, spermatogonia appeared normal and infertility was due to a

postnatal block in meiosis (Iwamori et al., 2012, PLoS One). We do think it is a valid point of dialogue and therefore included a new paragraph in the discussion addressing this point.

Reviewer Comment 5: From tSNE plots of single cell RNA-Seq data, it is concluded that there are opposing gradients in expression of stem and progenitor genes in germ cells at fetal and postnatal stages of development (Fig. 4 and supplementary Fig. 4). However, from the data shown this is not always evident. For instance, while gradients in SSC genes *Lhx1*, *Etv5* and *Id4* are apparent at E16.5, P0 and P6, they appear uniformly expressed in germ cells at P3. *Ret* also seems uniformly expressed in P3 germ cells while at E16.5 appears expressed at low levels in few germ cells (Supplementary Fig. 4). Given that *Gfra1* is the co-receptor for GDNF with *Ret* and *Gfra1* is associated with SSCs, the authors should show expression of *Gfra1* at all timepoints by tSNE plot. Gradients in expression of progenitor and differentiation genes are also not always evident at all timepoints. *Sohlh1* appears uniformly expressed at P0 and P3 while *Stra8* expression peaks in a population toward the middle of the tSNE plot at p6 (Fig. 4). The authors should confirm whether these markers are heterogeneously expressed as proposed at these timepoints by other analyses or other means of data presentation. For some of these genes, it should be confirmed that protein levels follow changes in levels of the mRNAs as the two may not always correlate. Additionally, the group of Yumiko Saga has previously reported that prospermatogonia are heterogeneous for expression of some SSC-linked genes in the fetal testis (Pui and Saga, Mech Dev 2017, 144:125-139). This paper should be quoted and results discussed in relation to this report.

Author Response: We would like to thank the reviewer for acknowledging the deficiencies in our presentation of the transcriptional gradients. Previous scRNA-seq studies of the male germline have reported transcriptional gradients or continuums, but did so in a more straightforward manner which was taken into account in revising the text. We have made changes within the text and the figures to address this issue. Briefly, we now refer to the transcriptional gradients as “continuums” rather than gradients to avoid confusion with physical gradients such as growth factor gradients. Also, we believe that tSNE representation of these transcriptional continuums is somewhat subjective based on the reader. Furthermore, the relative distribution of each sub-population within each age adds complexity to the data representation. For example, the more uniform appearance of SSC-associated markers at P3 that was noted likely reflects that ~70% of the population at P3 consists of ID4-eGFP^{Bright} SSCs (based on FCA quantifications in Fig. 2). Therefore, to provide a clearer representation of the data, we have now computed a one-dimensional tSNE distribution (or a continuum) for all cells within each developmental age. Then, we plotted the expression of several markers, including *Gfra1*, in heatmaps to better represent the gene expression pattern we were describing. Furthermore, the Pui and Saga (2017) study has now been referenced and discussed. Finally, rationale for marker selection has been incorporated within the text and gene expression data for additional marker genes is provided in supplemental information.

Reviewer Comment 6: Clustering analysis of single cell RNA-Seq data shown in Figure 4d seems counter-intuitive – why are cells in clusters 7, 8 and 10 split into multiple populations in the tSNE plot? For instance, a subset of cells from cluster 7 is found in the middle of the cluster 8 population and a fraction of cluster 10 cells are in the cluster 7 population. The authors should try other clustering methods, e.g. SC3 (Kiselev et al., Nature 2017). Changing parameters in the current algorithm may also improve clustering analysis. A dendrogram should be provided to help in visualizing cell clusters and discriminating genes. Key genes distinguishing clusters in Figure 4e should be displayed. Additionally, pre-filter thresholds used in the analysis (page 21 of Methods) are very different between developmental timepoints and even between replicates of the same timepoint. For instance, 70,000 UMI counts for P3 replicate #1 and 35,000 UMI counts for P3 replicate #2. Why is this? Appropriate quality control graphs for the datasets should be included in supplementary figures. The cut-off for low quality transcriptomes based on mitochondrial gene counts (25%) seems high – 10% is typically used.

Only one animal is used for the P6 timepoint while 2-3 animals are used for other timepoints. This may limit robustness of the P6 dataset due to sample variation. Ideally, 2 biological replicates should be used for each timepoint.

Author Response: We would like to thank the reviewer for identifying areas of concern with our scRNA-seq analysis. As downstream analyses build upon each other in our study, our choice of analyses must be clearly described and rooted in solid rationale. The appropriate information has been included in the text and is addressed as follows.

It is interesting that clusters 7, 8, and 10 split into multiple clusters as visualized by tSNE. Upon closer examination, we noticed that the certain factors are underlying these occurrences. For example, cluster 10 divided into two separate clusters based on developmental age (the smaller portion of cluster 10 present within the P3 dataset and the larger present within the P6 dataset). Cluster 8 behaved similarly, which we think caused the subdivision of cluster 7. Therefore, we believe the issue lies with the use of tSNE to visualize such a complex dataset. Indeed, recent computational studies have illustrated that UMAP (Uniform Manifold Approximation and Projection) is more appropriate for visualization of cellular relationships within scRNA-seq datasets that possess both local and global complexities, where tSNE is more appropriate for local complexities only (Becht et al., 2018, Nat Biotech). In our case, local substructure consisted of each developmental age and global structure encompassed relationships across the entire developmental window analyzed. In the revised Fig. 4d, under the same clustering parameters, the division of clusters 7, 8, and 10 are no longer present when visualized by UMAP. Also, we reanalyzed the individual ages by UMAP and noticed no change in the overall graphing visualization; therefore, tSNE was retained for those visualizations. Additionally, we have included a UMAP indicating the developmental ages (Supplemental Fig. 4a) as we felt this to be valuable information lacking in the previous version of the manuscript.

In terms of clustering method, we appreciate the reviewer's suggestion of additional algorithms to employ; we have attempted to use several different ones. However, the Louvain implementation with Seurat provided what we felt is the best representation of the germ cell subsets based on functional data provided by our current study and previously published studies. Indeed, while SC3 is a reliable method of clustering for many studies, the developers of SC3 note its limitation in clustering an excess of 5,000 cells and poor representation of rare populations such as stem cells (Kiselev et al., 2017, Nat Methods). For these reasons, we felt that SC3 is not suitable for our current study.

As suggested by the reviewer, a dendrogram is now included with the heatmap present in Fig. 4c. Also, because gene names were too small to see with the size of the heatmap, a table of the genes that define each cluster are included as new Supplementary Table 1.

Filtering parameters based on UMI count was done to exclude (1) a handful of doublets that were present within each dataset or (2) low-quality cells with very low UMI counts. Doublets were easily identified as cells with exceedingly high gene and UMI counts compared to the remaining population within each replicate. Filtering based on percent mitochondrial genes was performed across all replicates equally and conservatively for the following reasons: (1) previous studies by our group utilizing the same method of cell isolation and sorting resulted in limited cell death; (2) mitochondrial and/or apoptotic genes were not overtly represented in the dataset; and (3) the level of mitochondrial gene expression is heavily dependent upon cell type, proliferative status, and developmental state. Thus, we believe a conservative approach to filtering based on mitochondrial gene expression was appropriate for the current study. The appropriate explanations and rationale for filtering are now included in the revised Methods section.

Reviewer Comment 7: One of the most interesting parts of the manuscript is pseudotime analysis of SSC clusters at different developmental timepoints (Fig. 5) as this might reveal regulators of SSC development. This component could be explored/discussed further. Can authors confirm by e.g. immunostaining, that some of the genes of interest are dynamically expressed during conversion of prospermatogonia to SSCs? Is the pattern of Hox gene expression in developing SSCs expected? It is

mentioned that germ cells from P3 and P6 overlap in the predicted development timeline, suggesting that SSCs are defined by P3. However, there also appears large overlap between cells at P0 and P3 (Fig. 5a) so are SSCs be established earlier?

Author Response: We agree that tremendous insights can be gleaned from the pseudotime analysis of clusters associated with SSCs and their precursors. We have provided a list of these genes in the new Supplementary Table 2 so that future investigations in the field can be spurred from our findings. For the present study, use of immunostaining was generally avoided for two reasons: (1) reproducibility of immunostaining as a quantitative approach is challenging due to differential binding efficiencies of antibodies, and (2) previous studies in the field have attempted to elucidate the complexities of SSC specification and/or establishment utilizing antibody-based approaches with limited and mixed results. One explanation for the latter is that marker gene expression is highly dynamic during development. As the reviewer notes in comment #5, some associated markers have more or less restrictive expression for the given cell fate depending on the developmental age. Furthermore, it is important to note that maturation of cell lineages occurs asynchronously; thus, overlap among the developmental ages in pseudotime reflects differences in the state of each cell at each developmental age. Overlap between P0 and P3 likely reflects this, as it does for other developmental ages as well. Importantly, we conclude that establishment of the SSC population completes by P3 based first on flow cytometry data that illustrates a consistent number of ID4-eGFP^{Bright} spermatogonia from P3 to adulthood. This conclusion is supported by the scRNA-seq data. These viewpoints and interpretations are reflected in the revised text of the discussion section.

Reviewer Comment 8: As a minor point - are the authors confident that all germ cells are lineage-marked by the Blimp1-Cre transgene in their model system? This may influence the single cell results and interpretation.

Author Response: We agree with the reviewer that the extent of recombination is an important consideration. For the current study, we calculated recombination efficiency at >98% based on the percentage of neonatal ID4-eGFP+ spermatogonia that were also tdTomato+. Also, previous studies report that Blimp1-Cre driven recombination in PGCs is highly efficient (Ohinata et al., 2005, Nature). Thus, we are confident that utilization of Blimp1-Cre and Rosa(26)-tdTomato effectively labeled most, if not all, of the germline.

Response to Reviewer #2 Comments

Reviewer Comment 1: Fig. 4h: Sox2 is classified as pluripotency marker while Sox3 is classified as progenitor marker. However, Sox2 and Sox3 are functional similar. For example, a study has shown that "...ectopic expression of SOX2 in the testes functionally rescues the spermatogenic defect of Sox3 null mice." (<https://www.ncbi.nlm.nih.gov/pubmed/28515211>). Interestingly, the Sox2 and Sox3 seem to be negatively correlated based on Fig. 4h.

Author Response: It is interesting that Sox2 and Sox3 are negatively correlated in the mammalian male germline, but are functionally similar. While the mentioned publication illustrates interesting functional redundancies in Sox2/3 proteins, regulatory elements upstream and downstream the Sox3 locus were retained in that study. Thus, temporal regulation of the ORF is preserved regardless of whether Sox2 or Sox3 are expressed from the locus. In our study, the temporal expression from that locus was merely used as a marker for cell type identification. Our selection of marker genes, including Sox3, was based on functional studies that verify the gene requirement in either SSC maintenance or differentiation in the male germline. These views are now reflected in the revised text.

Reviewer Comment 2: What is the definition of percentage of expression (what is the cutoff to define expressed genes) in Fig. 4h?

Author Response: We thank the reviewer for identifying this unclear explanation. The appropriate corrections have been made in the text, figure legends, and figure images. For reference, “percent of expression” (renamed “percent of cells with detectable expression”) is intended to describe the percentage of cells where transcripts for the gene of interest were detected above signal noise.

Reviewer Comment 3: Figure 4: When pooling different ages/mice into one analysis, do you investigate possible batch effects?

Author Response: We agree that this is an important analysis that was absent in the initial version of the manuscript. In the revised Results section, we have included outcomes from tests of multivariate Pearson’s correlations which reflect negligible batch effects between replicates.

Reviewer Comment 4: For GO and Pathway analysis, what are the background genes (control genes)? Do you use all the genes or expressed genes as background?

Author Response: We thank the reviewer for acknowledging that this information was absent in the original version. The appropriate information has now been included in the revised Methods section. For reference, a list of detected genes from the entire scRNA-seq library was used as background for each gene-set enrichment analysis.

Reviewer Comment 5: Page 11: For the 10 clusters identified, several marker genes can clearly characterize the function of each cluster. But how do you choose marker genes? For example, in the manuscript (Fig. 4h), there are three pluripotency markers (Dppa3, Nanog and Sox2). However, there are many other well-known pluripotency markers, such as Oct-4, Ssea1. Are they not expressed or not selected?

Author Response: We agree that rationale for the selection of markers genes is important to discuss and necessary for the overall interpretation of the data. The revised Results section includes rationale for selection of markers and data representation of additional markers is in the new Supplementary Fig. 4. In terms of pluripotency genes presented in Fig. 4 and Fig. 1, the selected markers represent a gene set within the germline that are down-regulated prior to birth in mammals and associated with the loss of pluripotent state that exists in primordial germ cells (Reik and Surani et. al. 2015). Oct4(Pou5f1) is included with the analysis of E16.5 in Fig. 1.

Response to Reviewer #3 Comments

Reviewer Comment 1: I do not know what definition of an asymmetrical division the authors use. For me it means that already at division the daughter cells of a stem cell are different and will develop in different directions. For this to establish, the authors would have to show differences in the daughter cells already at the time of their formation by division from their mother cell. Clearly, the authors do not provide any evidence for this to occur, like it was shown for example in Drosophila where the daughter cells remaining in contact with the hub cells are different from those pushed away from it, already during division. The daughter cells may just as well be similar to each other and develop differently later on, depending on whether or not the local environment is inductive towards differentiation. The authors should delete any reference about symmetric or asymmetric divisions. For example, in page 7 lines 7-10: It should read that gradually more of the SSC daughter cells were induced to differentiate.

Author Response: We would like to thank the reviewer for addressing this issue and agree that our conclusions regarding asymmetrical/symmetrical division were over interpreted from the dataset in hand and additional evidence is needed to draw a firm conclusion. Statements regarding symmetry of cell division dynamics have been tempered throughout the text.

Reviewer Comment 2: Page 2, line 14: Sex determination takes place at about E12.5 and as the prespermatogonia become quiescent at E16.5, I would not call that “Shortly”.

Author Response: We agree that E12.5 to E16.5 is not considered “shortly” in the context of development time in mice and have revised the text accordingly.

Reviewer Comment 3: Page 8, lines 4-6: I do not understand the last part of this sentence, when cells are equally proliferative and clearly do proliferate why add “but not necessarily quiescent”? This should be clarified.

Author Response: We thank the reviewer for pointing out the lack of clarity in this statement and have revised the text accordingly.

Reviewers' comments:

Reviewer #1 (Remarks to the Author):

This revised manuscript by Law et al., has addressed some key concerns and is significantly improved. In particular, the single cell RNA-Seq analysis is more refined and the unsupported conclusions regarding symmetric and asymmetric division of SSCs removed. The presented data could be a useful resource for the field. However, as detailed below, multiple points raised from the first submission have not been satisfactorily addressed and key observations have not been investigated in sufficient depth. As such, additional data is still required to support the authors conclusions and provided model.

Major points:

1. In order to assess fate of different populations of fetal germ cells by functional assay the authors transplanted ID4 positive and negative E18.5 germ cells and found that only the minor ID4+ fraction generated colonies. It is concluded that the data supports a model whereby SSC fate is preprogrammed in a subset of fetal germ cells (potentially those with better genomic integrity) prior to postnatal testis development. The ability to generate spermatogenic colonies from fetal germ cells when transplanted into an adult recipient testis is surprising. Studies from the Griswold lab have shown that the ability to form spermatogenic colonies in this setting is only possible once the SSC pool has fully developed postnatally from day 4 after birth (McLean et al., *Biology of Reproduction* 2003, 69:2085-2091). Interestingly, the one donor colony shown in the manuscript (Fig. 6b) appears quite small, suggesting inefficient repopulation. Do the donor colonies successfully establish spermatogenesis and would this assay be the most appropriate from fetal donors? Despite comments by the authors, lineage tracing to define fate should be possible as they have identified in their single cell analysis genes that are restricted to the ID4 negative population at these fetal stages (Sohlh1, Sox3 etc.). Further, as by birth the large majority of germ cells have become ID4 positive in the absence of cell division (Fig. 1e), does this not suggest that almost all fetal germ cells might have the capacity to become SSCs and that SSC fate is not preprogrammed in a subset of fetal cells?

2. In response to comments, the authors have removed mention of symmetric and asymmetric SSC division from the manuscript as the available data did not support this concept. However, this point was one major aspect of novelty in the original manuscript so it is disappointing that the authors chose not to provide additional data in support of their original model. This limits the advance of the manuscript.

3. Comparison of stem cell capacity of ID4-Bright and Mid populations was requested as the authors provide a model whereby functional capacity of these populations is distinct although has not been defined in detail previously. As highlighted by the authors, in a recent publication they show in an ex vivo experiment that ID4-Bright cells are more resistant to the differentiation stimulus retinoic acid (RA) than ID4-Mid cells and these populations should therefore have different fate (Lord et al., *Stem Cell Reports* 2018). While this suggests that SSC potential of these populations is distinct, this needs to be formally demonstrated through transplantation. The authors state that this was not feasible as ID4-Bright and ID4-Mid cells could not be distinguished effectively by FACS. However, if this was the case, how was the response of ID4-Bright and ID4-Mid cells to RA defined? Further, it was originally requested that the authors assess whether the ID4-Bright/Mid/Dim populations in postnatal testis are positive for differentiation marker c-Kit. This could be tested by flow cytometry very simply but has not been attempted. This would aid in characterizing germ cell populations in developing testis.

4. Further characterization of germ cell nests in the developing testis was requested as data were intriguing but rather preliminary (Fig. 3). However, higher magnification images of the nests have not been provided as requested and assessment of the syncytial nature not attempted as

suggested despite availability of antibodies to intercellular bridge components. Of note, clusters of germ cells in the embryonic testis have already been described (Mork et al., Mech Dev 2012, 128:591-596). Can this explain the existence of germ cell nests in postnatal testis? As discussed by the authors, although loss of intercellular bridge components does not seemingly disrupt SSC formation, the single cell vs. syncytial state is suggested to be important for SSC identity so the distinct ID4 populations might be expected to have different topologies.

5. Expression of some novel genes identified in the single cell RNA-Seq analysis of germ cells at different developmental timepoints was requested to be confirmed by immunostaining. It is disappointing that the authors have not attempted this as it would provide a more thorough analysis. It is argued that there are limitations to immunostaining analysis, which is correct, but it could still provide very effective semi-quantitative data documenting changes in levels of candidate proteins in the SSC population during testis development.

Minor point: UMAP visualization of single cell RNA-Seq data is stated to be included as Suppl Fig. 4a. However, the provided supplementary figures do not seem to contain this panel.

Reviewer #2 (Remarks to the Author):

The authors have addressed all of my concerns and questions.

We are again grateful for the valuable insight by the reviewer and have tried to address the concerns with changes to the text, addition of new data, or by providing rationale for why a revision was not made. Revisions from the first round of review are highlighted as yellow and those from the second round of review are highlighted as blue.

Reviewer Comment: In order to assess fate of different populations of fetal germ cells by functional assay the authors transplanted ID4 positive and negative E18.5 germ cells and found that only the minor ID4+ fraction generated colonies. It is concluded that the data supports a model whereby SSC fate is preprogrammed in a subset of fetal germ cells (potentially those with better genomic integrity) prior to postnatal testis development. The ability to generate spermatogenic colonies from fetal germ cells when transplanted into an adult recipient testis is surprising. Studies from the Griswold lab have shown that the ability to form spermatogenic colonies in this setting is only possible once the SSC pool has fully developed postnatally from day 4 after birth (McLean et al., *Biology of Reproduction* 2003, 69:2085-2091). Interestingly, the one donor colony shown in the manuscript (Fig. 6b) appears quite small, suggesting inefficient repopulation. Do the donor colonies successfully establish spermatogenesis and would this assay be the most appropriate from fetal donors?

Author Response: We appreciate the reviewer's perspective on the transplant results but respectfully disagree. Although the McLean et al., 2003, BOR study indicated that SSC engraftment in adult recipient testes is only possible with donor cells post P4, a study by the Brinster lab (Kubota et al., 2004, *Biol. Repro.* 71: 722-31) demonstrated that this is not accurate. In the Kubota et al. study, germ cells isolated from P0.5-1.5 donor mice were shown to engraft in adult recipient testes (see Figure 5 of that paper). Regarding the colony shown in Figure 7b (previously Figure 6b) of our manuscript, from our perspective the colony is not quite small. Based on the thousands of colonies generated by transplant that we have examined over the years the colony is of average size. A scale bar has been added to the figure so that readers can make their own judgments of the size of the colony and relate it to those in previously published studies.

We believe that the transplant results demonstrate that a defined subset of the prospermatogonial population at E18.5 in mice (i.e. those marked as ID4-eGFP+) is endowed with the capacity to function as stem cells. This is a simple yes or no experiment; either the ID4-eGFP+ and/or ID4-eGFP- prospermatogonia produce colonies after transplant or they do not. If they do, a functional definition of a stem cells is met. We transplanted nearly 10-times more ID4-eGFP- than ID4-eGFP+ prospermatogonia and did not observe any colonies. From our viewpoint, this outcome is unequivocal and supports our conclusion that SSC fate is established in a defined subset of prospermatogonia as early as E18.5.

Reviewer Comment: Despite comments by the authors, lineage tracing to define fate should be possible as they have identified in their single cell analysis genes that are restricted to the ID4 negative population at these fetal stages (Sohlh1, Sox3 etc.). Further, as by birth the large majority of germ cells have become ID4 positive in the absence of cell division (Fig. 1e), does this not suggest that almost all fetal germ cells might have the capacity to become SSCs and that SSC fate is not preprogrammed in a subset of fetal cells?

Author Response: We understand the reviewer's perspective on lineage tracing and agree this assessment would complement the transplantation results. However, we disagree that the experimental approach is feasible at this time. Although some genes such as those pointed out by the reviewer may

be uniquely expressed by the ID4 negative germ cells in fetal development, an inducible Cre transgene for a candidate would be required to assess whether any of the ID4 negative prospermatogonia give rise to SSCs in postnatal development. For the genes pointed out by the reviewer (Sohlh1 and Sox3), inducible Cre transgene models are not currently available through a repository. Thus, we would need to produce new lines from scratch, a process that would take 1-2 years to achieve. All in all, obtaining sufficient resources to carry out the lineage tracing for ID4 negative prospermatogonia would require a 2-3 year timeframe. From our perspective, the lineage tracing would simply confirm what transplantation has already shown, thus going down this path is not feasible at present or necessary to draw conclusions. Note that use of a non-inducible Cre transgene would not work if the gene is turned on again at any stage in postnatal spermatogenesis, and this is the case for both Sohlh1 and Sox3.

Regarding the reviewer's assertion that a large majority of germ cells have become ID4 positive by birth in the absence of cell division and this suggests that almost all fetal germ cells might have capacity to become SSCs, we respectfully disagree that this is what the data indicates. First, ~45% of the germ cell population is ID4 positive at birth; from our perspective this is not a large majority. Second, there is an increase in proliferative index of the ID4 positive population at P0, raising from ~2% at E18.5 to ~15% at P0. Based on these data, we believe that the increase in ID4 positive prospermatogonial number from ~2,000 at E18.5 to ~4,000 at P0 can be attributed to a rise in proliferation index. We have modified the text in various parts of the manuscript as well as the model presented in Figure 7 (previously Figure 6) to align more closely with these findings.

Reviewer Comment: In response to comments, the authors have removed mention of symmetric and asymmetric SSC division from the manuscript as the available data did not support this concept. However, this point was one major aspect of novelty in the original manuscript so it is disappointing that the authors chose not to provide additional data in support of their original model. This limits the advance of the manuscript.

Author Response: We agree that the concept of symmetric vs asymmetric division in the mammalian male germline is important to explore. However, at this time we cannot devise an experimental strategy that addresses this directly. Regardless, we appreciate the reviewer's opinion that without additional data to support the concept the advance of the manuscript is limited, but respectfully disagree. There are a multitude of important advances made by this study and the merit overall does not rest with any single aspect especially a concept for which direct experimental evidence is difficult to obtain. Indeed, exploring the symmetry of cell division by prospermatogonia/spermatogonia is an entire study in itself and we anticipate that the findings in our current manuscript will entice others to study this specific aspect of male germline stem cell development.

Reviewer Comment: Comparison of stem cell capacity of ID4-Bright and Mid populations was requested as the authors provide a model whereby functional capacity of these populations is distinct although has not been defined in detail previously. As highlighted by the authors, in a recent publication they show in an ex vivo experiment that ID4-Bright cells are more resistant to the differentiation stimulus retinoic acid (RA) than ID4-Mid cells and these populations should therefore have different fate (Lord et al., Stem Cell Reports 2018). While this suggests that SSC potential of these populations is distinct, this needs to be formally demonstrated through transplantation. The authors state that this was not feasible as ID4-

Bright and ID4-Mid cells could not be distinguished effectively by FACS. However, if this was the case, how was the response of ID4-Bright and ID4-Mid cells to RA defined?

Author Response: We respectfully disagree with the reviewer's perspective that our proposed model indicates that the Mid and Bright populations have different functional capacities. Our models presented in Figure 1a and Figure 7 (previously Figure 6) as well as models we have presented in other publications (Helsel et al., 2017; Lord et al., 2018) imply that the Mid population is in transition to a progenitor state and can revert back to a Bright/SSC state. These models account for the possible capacity of Mid cells to function as SSCs in certain contexts similar to the functional capacity of Bright cells. We have modified the arrows in the models presented in Figure 1a and Figure 7 (previously Figure 6) to more closely reflect the concept that Mid cells are able to revert back to Bright cells and function as SSCs. We have also revised the text in multiple places to more accurately reflect these viewpoints that are supported by our datasets.

Our previous response to the reviewer's critique about lack of functional assessment for Mid cells was due to technical limitations of not being able to obtain a pure population via FACS cell sorting. Again, clean isolation of the Mid cells from the Bright cells via FACS for downstream analyses is not possible, at least in our hands. Thus, carrying out functional transplantation analyses is not feasible. Measuring a response to RA via attainment of c-Kit expression does not require cell isolation to obtain populations for transplantation. Assessment with a scatter plot can distinguish a c-Kit phenotype in the different populations. While flow cytometer is used for both types of analysis, they are not identical approaches on a technical level. From a cell sorting standpoint, purification of Bright and Mid cells for transplantation has not been possible in our hands. The Mid population always contains some Bright cells. Although we could apply more stringent gating parameters, to remove all Bright cells, this strategy would also eliminate a portion of the Mid population and therefore reduce rigor of any data generated from downstream analysis. For these reasons, we prefer to err on the side of caution and propose models that leave open the possibility that Mid and Bright cells have similar functional capacities which has been done in the current study.

Reviewer Comment: Further, it was originally requested that the authors assess whether the ID4-Bright/Mid/Dim populations in postnatal testis are positive for differentiation marker c-Kit. This could be tested by flow cytometry very simply but has not been attempted. This would aid in characterizing germ cell populations in developing testis.

Author Response: We recognize the reviewer's previous comment and apologize for not responding directly. The reason for not doing so is that information already exists in the literature to address the reviewer's question, at least in part. In particular, Lord et al. 2018, Stem Cell Reports showed that a large portion of the Dim population is c-Kit+ whereas as less than 50% of Mid cells are c-Kit+ and the Bright cells are c-Kit- in testes of P6 mice. In addition, previous studies by Ohbo et al., 2003, Dev. Biol. indicate that prospermatogonia in P0 mice are c-Kit-. Because information already exists in the literature for c-Kit expression by the germ cell population of P0 and 6 mouse testes, we did not feel that repeating these assessments in the current study would add much value. However, we appreciate the reviewer's interest and have included a new Supplemental Figure 4e of the c-Kit profiles in ID4 Bright/Mid/Dim/Negative populations at P3, the age point in development where our model predicts the SSC pool is established and other spermatogonial subsets begin to develop. The data are in with the gene expression profiles generated by our scRNAseq analyses.

Reviewer Comment: Further characterization of germ cell nests in the developing testis was requested as data were intriguing but rather preliminary (Fig. 3). However, higher magnification images of the nests have not been provided as requested and assessment of the syncytial nature not attempted as suggested despite availability of antibodies to intercellular bridge components. Of note, clusters of germ cells in the embryonic testis have already been described (Mork et al., Mech Dev 2012, 128:591-596). Can this explain the existence of germ cell nests in postnatal testis? As discussed by the authors, although loss of intercellular bridge components does not seemingly disrupt SSC formation, the single cell vs. syncytial state is suggested to be important for SSC identity so the distinct ID4 populations might be expected to have different topologies.

Author Response: Although the Mork et al. study described clusters of germ cells in the fetal testis, the assessment did not extend beyond E13.5 which is a period in development when prospermatogonia are still mitotically active and genes that seemingly align with fate determination such as *Id4* are not yet expressed. Thus, we have referenced the study in the discussion of our manuscript as it relates to nest development in the male germline. However, we are hesitant to make predictions of nest formation having origins prior to prospermatogonial quiescence and induction of ID4 expression because data supporting this are lacking at the current time. Indeed, we feel that extensive future experimentation is required to fully address this possibility which is beyond the scope of the current study.

Regarding the interconnectedness of nested germ cells in the developing testis, we have not performed Text14 immunostaining because a body of knowledge regarding the intercellular bridge connections of prospermatogonia already exists in the literature. In particular, studies of Greenbaum et al., 2009, Biol. Reprod. as well as Lei and Spradling, 2013, Development indicate that in testes of P0 mice some prospermatogonia are linked by an intercellular bridge but other prospermatogonia that are present in close proximity are not linked. In addition, studies by Zamboni and Merchant, 1973, Am J Anat indicate that some prospermatogonia in fetal testes are linked by intercellular bridges and the high magnification image we provided in Figure 1d of a seminiferous cord at E16.5 shows three ID4-eGFP+ prospermatogonia in close proximity with two of the cells close enough to likely be linked by an intercellular bridge but a third cell that is not linked as a syncytium. Notably, the proximity of the three cells falls within the parameters we used for defining nests. Taken together, the body of knowledge that already exists in the literature and our findings indicate that syncytial connection is not associated with nest formation. We have added several sentences to the discussion section to reflect these interpretations and have referenced previous studies exploring interconnectedness of prospermatogonia. Note that although we feel that conducting Tex14 immunostaining in our studies is not warranted because of the existing knowledge base, we did attempt to conduct this in order to address the reviewer's comment. However, we were unable to make the commercially available antibodies work while preserving live eGFP and TdTomato fluorescence. Regardless, we feel that the observations made from the imaging provided and relating our findings to prospermatogonial interconnections reported in previous studies allows for valid interpretations to be made.

Reviewer Comment: Expression of some novel genes identified in the single cell RNA-Seq analysis of germ cells at different developmental timepoints was requested to be confirmed by immunostaining. It is disappointing that the authors have not attempted this as it would provide a more thorough analysis. It is argued that there are limitations to immunostaining analysis, which is correct, but it could still

provide very effective semi-quantitative data documenting changes in levels of candidate proteins in the SSC population during testis development.

Author Response: We appreciate the reviewer's perspective but disagree that rigorous comparisons can be made even at a semi-quantitative level using immunostaining. Regardless of the difference in opinion, we have attempted to address the reviewer's request by choosing genes that are differentially expressed at E18.5, P0, and P3 (one gene at each developmental age point) in a subset of germ cells that the trajectory analysis of scRNA-seq datasets predicts are fated to become SSCs. For each of these, immunostaining of cross-sections revealed co-localization in a subset of the germ cell population that appear to have high ID4-eGFP expression, thus validating the scRNA-seq analyses at the protein level. These data have been incorporated as a new Figure 5.

Reviewer Comment: UMAP visualization of single cell RNA-Seq data is stated to be included as Suppl Fig. 4a. However, the provided supplementary figures do not seem to contain this panel.

Author Response: We have now added the missing UMAP visual.

REVIEWERS' COMMENTS:

Reviewer #1 (Remarks to the Author):

The revised manuscript by Law et al. has addressed some key points and has been improved. The presented data will be of interest to the field. However, substantial inconsistencies between provided data, arguments made by the authors (main text and rebuttal) and the presented model (Fig. 7d) need to be resolved (see below). Some data provided in the revised manuscript is also preliminary and should be revised.

1. The evident controversy between studies from the Griswold and Brinster groups concerning the ability of fetal germ cells to successfully transplant into adult recipients should be discussed in the text. The Griswold study is often quoted and it would be useful for the field if this inconsistency is highlighted in reference to the authors transplantation data.

2. Within the stem cell field, it is known that transplantation and lineage-tracing approaches have distinct advantages and disadvantages in assaying stem cell activity (Snippert & Clevers EMBO Rep 2011 12:113). Lineage-tracing is the standard method used to define cell fate in intact tissue while transplantation assays place cell populations outside their normal physiological context and can provide mis-leading results. Dissociated fetal germ cells would never normally be present within the lumen of seminiferous tubules of adults as occurs during this transplantation procedure. It is appreciated that lineage-tracing studies are beyond the scope of the study but it should be highlighted in the text that definitive assessment of fetal cell fate requires confirmation by lineage-tracing approaches.

3. A central feature of the model presented by the authors is that a small subset of germ cells marked by Id4 expression at late stages of fetal development is preprogrammed to become the foundational stem cell pool (Fig. 7d). That a large fraction of germ cells becomes Id4-positive at birth (~45%) and by postnatal day 1 (~70%) (Fig. 1g) while proliferation rates are limited (Fig. 1f) suggests that most germ cells ultimately activate Id4 expression during testis development and could have stem cell potential, contrasting with this "preprogrammed fate" model. In the rebuttal the authors suggest that low but increasing proliferation rates of the Id4-positive fetal germ cells during development would account for this observation – the Id4-positive population that is preprogrammed for stem cell fate expands from late fetal stages to ultimately constitute a large fraction of total germ cells in the neonatal testis. However, it is shown that the proliferation rate of Id4-positive and Id4-negative germ cells is largely comparable from late fetal to neonatal stages (Fig. 1f). Therefore, if both these populations are proliferating (and in the absence of cell loss by e.g. apoptosis), wouldn't the overall number of germ cells increase during development but the relative proportions of Id4-positive and Id4-negative cells remain the same? This data would be more consistent with the alternative explanation that from late fetal to neonatal stages a large proportion of the germ cells activate Id4 expression. Confusingly, the authors also elude to this within the text on page 9: "From P0 to P2, the size of ID4-eGFPBright nests remained relatively constant (Fig. 3c), but the total number of nests steadily increased (Fig. 3d, inset), suggesting that germ cells gained ID4 expression in localized regions along seminiferous tubules (depicted in Fig. 3e, identified with "1")." These apparent inconsistencies between data and model should be resolved.

4. Due to technical limitations in FACS sorting, the authors have been unable to isolate pure populations of Id4-Bright and Id4-Mid cells to provide functional comparison of stem cell potential in support of their model. This is surprising given that the Id4-Bright and Mid populations exhibit a log-fold difference in fluorescence intensity (Helsel et al., Development 2017, Fig. 1c). The authors therefore propose that Id4-Bright and Mid cells can have similar stem cell capacities but that the Mid cells are transitioning out of the stem cell pool. While this proposal would be consistent with the role of Id4 in promoting the stem cell state, it is confusing that in the model provided (Fig. 7d), a subset of Id4-negative fetal germ cells is indicated to become Id4-Mid in the neonatal testis

but to differentiate with no possibility to contribute to the stem cell pool despite having similar functional potential as the Id4-bright cells. As discussed by the authors in the rebuttal, in P6 mice, 50% of Id4-Mid cells and most Id4-dim cells are c-Kit positive, an established marker of differentiating spermatogonia. Therefore, a substantial proportion of Id4-mid and dim cells are differentiated and are unlikely to be cells transitioning out of the stem cell pool as indicated in the model. These inconsistencies should be resolved. Also, what evidence do the authors have to conclude that both the first and second rounds of spermatogenesis originate from the Id4-negative fetal germ cell fraction (Fig. 7d)?

5. It is appreciated that additional characterization of the nests of germ cells was not possible due to technical limitations. It would therefore help the reader if the authors explained in the methods their criteria for a nest of germ cells, stated as 3 or more germ cells in proximity. Why was this number of cells selected and are the cells in a nest always in physical contact?

6. Immunofluorescence images are included in Figure 5 in order to confirm that candidate genes identified from the single cell RNA-seq analysis are altered at the protein level in different germ cell populations. These data look promising but preliminary as only single microscope images are shown with just a few germ cells per image. Are these images representative from multiple animals? At a minimum, additional microscope fields and samples from this analysis should be included or, preferably, appropriate quantification performed.

We thank the reviewer for eliciting an insightful debate about interpretations of the important discoveries made in this study and appreciate the opinions being shared. We have again tried to address the major concerns with revisions to the text of the manuscript. In particular, we have crafted entirely new paragraphs for the discussion section of the manuscript that present what we believe are more balanced interpretations of the data.

Reviewer Comment: The evident controversy between studies from the Griswold and Brinster groups concerning the ability of fetal germ cells to successfully transplant into adult recipients should be discussed in the text. The Griswold study is often quoted and it would be useful for the field if this inconsistency is highlighted in reference to the authors transplantation data.

Author Response: We have added the following sentence to the discussion section of the manuscript, *“While studies by McLean et al., 2004 indicated that donor germ cells prior to P4 are unable to colonize adult recipient testes, studies of Kubota et al., 2004 demonstrated that a subset of the prospermatogonial population in testes at P0-1 can engraft in adult recipient testes and generate colonies of spermatogenesis, consistent with findings in the present study.”*

Reviewer Comment: Within the stem cell field, it is known that transplantation and lineage-tracing approaches have distinct advantages and disadvantages in assaying stem cell activity (Snippert & Clevers EMBO Rep 2011 12:113). Lineage-tracing is the standard method used to define cell fate in intact tissue while transplantation assays place cell populations outside their normal physiological context and can provide mis-leading results. Dissociated fetal germ cells would never normally be present within the lumen of seminiferous tubules of adults as occurs during this transplantation procedure. It is appreciated that lineage-tracing studies are beyond the scope of the study but it should be highlighted in the text that definitive assessment of fetal cell fate requires confirmation by lineage-tracing approaches.

Author Response: We appreciate that in organ systems that do not have a well refined transplantation assay, lineage tracing is the predominant tool for investigation of stem cell capacity. Although lineage tracing is indeed a powerful approach, for the mouse spermatogenic lineage transplantation has been the “gold standard” to assay for stem cell function for over two decades. From our viewpoint, introducing subsets of prospermatogonia into an adult testis microenvironment and asking whether or not they are capable of engrafting could be considered the ultimate test of preprogramming for attainment of spermatogenic stem cell capacity. Again, we transplanted ~10-times more ID4-eGFP- prospermatogonia than ID4-eGFP+ prospermatogonia but only observed engraftment by the ID4-eGFP+ subset. We believe these findings make a clear delineation of the functional capacity between the subpopulations, but we do appreciate the reviewer’s opinion and concerns. Thus, the following sentences have been added to the discussion section:

“In the present study, the observation that defined subsets of prospermatogonia can generate colonies of spermatogenesis following transplantation into adult testes indicates that the propensity for SSC fate is already in place within a portion of the population at E18.5. Although this experimental approach creates a situation that is anomalous as prospermatogonia will not be present in adult testes in a normal developmental context, the capacity to engraft after transplant could be viewed as the ultimate functional test of spermatogenic stem cell capacity.”

“Importantly, approximately ten-fold more ID4-eGFP- prospermatogonia than ID4-eGFP+ prospermatogonia isolated from E18.5 testes were transplanted and donor-derived colonies from the ID4-eGFP- subpopulation were not observed.”

Based on these collective observations, we propose a conceptual model in which the capacity to attain SSC fate during neonatal development is not stochastic but rather preprogrammed in a subset of prospermatogonia during fetal development with a different subset of fetal prospermatogonia programmed to become the first differentiating spermatogonia in neonatal development. This model is consistent with findings of Kluin and de Rooij (1981) that observed subsets of prospermatogonia in fetal testes possessing nuclear morphologies that resembled either undifferentiated or differentiating postnatal spermatogonial subtypes.”

“While intriguing, validation of this preprogramming concept will require future experimentation using complimentary approaches such as lineage tracing and ablation to link trajectory predictions of all fetal prospermatogonial subsets to functional fates in postnatal life.”

Reviewer Comment: A central feature of the model presented by the authors is that a small subset of germ cells marked by Id4 expression at late stages of fetal development is preprogrammed to become the foundational stem cell pool (Fig. 7d). That a large fraction of germ cells becomes Id4-positive at birth (~45%) and by postnatal day 1 (~70%) (Fig. 1g) while proliferation rates are limited (Fig. 1f) suggests that most germ cells ultimately activate Id4 expression during testis development and could have stem cell potential, contrasting with this “preprogrammed fate” model. In the rebuttal the authors suggest that low but increasing proliferation rates of the Id4-positive fetal germ cells during development would account for this observation – the Id4-positive population that is preprogrammed for stem cell fate expands from late fetal stages to ultimately constitute a large fraction of total germ cells in the neonatal testis. However, it is shown that the proliferation rate of Id4-positive and Id4-negative germ cells is largely comparable from late fetal to neonatal stages (Fig. 1f). Therefore, if both these populations are proliferating (and in the absence of cell loss by e.g. apoptosis), wouldn't the overall number of germ cells increase during development but the relative proportions of Id4-positive and Id4-negative cells remain the same? This data would be more consistent with the alternative explanation that from late fetal to neonatal stages a large proportion of the germ cells activate Id4 expression. Confusingly, the authors also elude to this within the text on page 9: “From P0 to P2, the size of ID4-eGFPBright nests remained relatively constant (Fig. 3c), but the total number of nests steadily increased (Fig. 3d, inset), suggesting that germ cells gained ID4 expression in localized regions along seminiferous tubules (depicted in Fig. 3e, identified with “1”).” These apparent inconsistencies between data and model should be resolved.

Author Response: We appreciate the reviewer's perspectives on the data but have a different interpretation and believe the difference in opinion is rooted in how the SSC pool is being considered. The reviewer seems to be considering the ID4-eGFP+ population in totality as being reflective of the SSC pool. However, our model and interpretations consider the ID4-eGFPBright population as the SSC pool which is ~10% of the total germ cell population at P0, rising to ~40% at P1, and peaking at ~70% at P3 before rapidly declining to <10% at P6 where it remains into adulthood. In accordance, the proliferative index of the ID4-eGFPBright population increases from P0-3 ahead of the other germ cell subsets. We believe the proposed model in Figure 7d and concluding statements made throughout the manuscript are in-line and supported

by the data if one considers the SSC pool to be represented as the ID4-eGFP^{Bright} population. Importantly, we have shown in previous studies that this population is endowed with potent stem cell capacity. We have modified statements in several places throughout the manuscript to reflect these interpretations, in particular is the second section of the results and first paragraph of the discussion.

Reviewer Comment: Due to technical limitations in FACS sorting, the authors have been unable to isolate pure populations of Id4-Bright and Id4-Mid cells to provide functional comparison of stem cell potential in support of their model. This is surprising given that the Id4-Bright and Mid populations exhibit a log-fold difference in fluorescence intensity (Helsel et al., Development 2017, Fig. 1c). The authors therefore propose that Id4-Bright and Mid cells can have similar stem cell capacities but that the Mid cells are transitioning out of the stem cell pool. While this proposal would be consistent with the role of Id4 in promoting the stem cell state, it is confusing that in the model provided (Fig. 7d), a subset of Id4-negative fetal germ cells is indicated to become Id4-Mid in the neonatal testis but to differentiate with no possibility to contribute to the stem cell pool despite having similar functional potential as the Id4-bright cells. As discussed by the authors in the rebuttal, in P6 mice, 50% of Id4-Mid cells and most Id4-dim cells are c-Kit positive, an established marker of differentiating spermatogonia. Therefore, a substantial proportion of Id4-mid and dim cells are differentiated and are unlikely to be cells transitioning out of the stem cell pool as indicated in the model. These inconsistencies should be resolved. Also, what evidence do the authors have to conclude that both the first and second rounds of spermatogenesis originate from the Id4-negative fetal germ cell fraction (Fig. 7d)?

Author Response: We appreciate the reviewer's viewpoints on the ID4-Mid and Dim populations but disagree that there are inconsistencies with the model we have proposed for Figure 7d. Regardless of the differences in opinion, we have attempted to address the concerns by including new paragraphs in the discussion section that reflect the points discussed below.

That the reviewer finds our inability to cleanly separate Bright and Mid populations surprising is appreciated but the fact is that we have not been unable to achieve this at a level we feel is rigorous enough for experimental testing by transplantation analyses.

We believe that the reviewer's confusion about how ID4-Mid cells could be arising from ID4-negative cells derives from a misunderstanding about the very different developmental context between the late fetal/early neonatal and more advanced postnatal period for which we originally described the Bright-Mid-Dim-Negative subsets. The ID4-Mid population described in the Helsel et al., 2017, Development study was for P8 testes which is a point in mouse development when the postnatal spermatogonial populations are fully established and therefore ID4-Mid cells arising from the Bright pool is logical. In the current study, we describe (as depicted in the model of Figure 7d) ID4-Mid cells arising at P0-3 prior to the ID4-Bright SSC pool being fully established. Thus, it is logical to propose that these cells arise from a subset of ID4-Negative prospermatogonia. Of course, this will need to be validated in future studies.

In regards to the comment that 50% of ID4-Mid cells and most ID4-Dim cells are c-KIT⁺ at P6 and therefore differentiated so unlikely to be cells transitioning out of the stem cell pool, we must respectfully disagree with this interpretation and believe that the reviewer is under a misconception about how c-KIT⁺ spermatogonia arise. First, the 50% of ID4-Mid and all ID4-Dim cells that are c-KIT⁺ at P6 could be derived from prospermatogonia that adopted this

trajectory directly from a prospermatogonial state and therefore did not arise from the SSC pool. Indeed this is what our proposed model implies and is supported by results of Kluin and de Rooij, 1981 that showed some prospermatogonia adopt a differentiating spermatogonial fate from the prospermatogonial stage directly. Second, essentially all postnatal undifferentiated progenitor spermatogonia (most Apair and Aaligned) are known to transition to a differentiating state (that would involve becoming c-KIT+) following retinoic acid exposure and this occurs in the absence of a cell division (see studies of Tegelenbosch and de Rooij, 1993). In other words, transit amplifying progenitor spermatogonia that are ID4-Mid and Dim would become c-KIT+ when induced to undergo the undifferentiated-to-differentiating transition by retinoic acid signaling. Thus, if the ID4-Mid and ID4-Dim cells present at P6 are derived from the SSC pool (i.e. transitioning out of the stem cell state), being c-KIT+ is to be expected.

In regards to what the evidence is to suggest that the first and second rounds of spermatogenesis arise from ID4-eGFP- germ cells as suggested by the model in Figure 7d, this is a proposed concept that we believe is supported circumstantially by the data. The transplantation analyses indicate that ID4-eGFP- fetal germ cells do not have the capacity to function as stem cells in a postnatal testicular environment and therefore by deduction are fated for non-SSC paths. Moreover, if the ID4-eGFP- (c-KIT+) cells present at P3 are not derived from the ID4-eGFP- fetal germ cells, they would therefore be derived from the ID4-eGFP+ pool which seems unlikely to us. However, we do appreciate the reviewer's arguments and have revised the discussion to include our rationale but also make clear that future experimentation is required to functionally test the postulations that we are making.

Reviewer Comment: It is appreciated that additional characterization of the nests of germ cells was not possible due to technical limitations. It would therefore help the reader if the authors explained in the methods their criteria for a nest of germ cells, stated as 3 or more germ cells in proximity. Why was this number of cells selected and are the cells in a nest always in physical contact?

Author Response: The choice of cell number to define a nest was based on a rationale that 2 cells could be in close proximity by chance or following a cell division but prior to migration away from each other and therefore would not constitute a nest. The results section has been revised to reflect this rationale. Again, defining physical contact of the nests is technically not feasible at this time and we believe beyond the scope of the current study.

Reviewer Comment: Immunofluorescence images are included in Figure 5 in order to confirm that candidate genes identified from the single cell RNA-seq analysis are altered at the protein level in different germ cell populations. These data look promising but preliminary as only single microscope images are shown with just a few germ cells per image. Are these images representative from multiple animals? At a minimum, additional microscope fields and samples from this analysis should be included or, preferably, appropriate quantification performed.

Author Response: We have included additional images as a new Supplementary Figure 5 and indicated within the legend for Figure 5 that images are representative of n=2-3 testes from different animals and 3-5 cross-sections for each developmental age point.